



# A new sampling capability for uncertainty quantification in the Ice-sheet and Sea-level System Model v4.19 using Gaussian Markov random fields

Kevin Bulthuis[1] and Eric Larour[1]

[1]Jet Propulsion Laboratory, California Institute of Technology, Pasadena, CA 91109, USA

**Correspondence:** Kevin Bulthuis (kevin.m.bulthuis@jpl.nasa.gov)

**Abstract.** Assessing the impact of uncertainties in ice-sheet models is a major and challenging issue that needs to be faced by the ice-sheet community to provide more robust and reliable model-based projections of ice-sheet mass balance. In recent years, uncertainty quantification (UQ) has been increasingly used to characterize and explore uncertainty in ice-sheet models and improve the robustness of their projections. A typical UQ analysis involves first the (probabilistic) characterization of the sources of uncertainty followed by the propagation and sensitivity analysis of these sources of uncertainty. Previous studies concerned with UQ in ice-sheet models have generally focused on the last two steps but paid relatively little attention to the preliminary and critical step of the characterization of uncertainty. Sources of uncertainty in ice-sheet models, like uncertainties in ice-sheet geometry or surface mass balance, typically vary in space and potentially in time. For that reason, they are more adequately described as spatio(-temporal) random fields, which account naturally for spatial (and temporal) correlation. As a means of improving the characterization of the sources of uncertainties in ice-sheet models, we propose in this paper to represent them as Gaussian random fields with Matérn covariance function. The class of Matérn covariance functions provides a flexible model able to capture statistical dependence between locations with different degrees of spatial correlation or smoothness properties. Samples from a Gaussian random field with Matérn covariance function can be generated efficiently by solving a certain stochastic partial differential equation. Discretization of this stochastic partial differential equation by the finite element method results in a sparse approximation known as a Gaussian Markov random field. We solve this equation efficiently using the finite element method within the Ice-sheet and Sea-level System Model (ISSM). In addition, spatio-temporal samples can be generated by combining an autoregressive temporal model and the Matérn field. The implementation is tested on a set of synthetic experiments to verify that it captures well the desired spatial and temporal correlations. Finally, we demonstrate the interest of this sampling capability in an illustration concerned with assessing the impact of various sources of uncertainties on the Pine Island Glacier, West Antarctica. We find that both larger spatial and temporal correlations lengths will likely result in increased uncertainty in the projections.



# 1 Introduction

Despite large improvements in ice-sheet modeling in recent years, model-based estimates of ice-sheet mass balance remain characterized by large uncertainty. The main sources of uncertainty are associated with limitations related to poorly modeled physical processes, the model resolution, poorly constrained initial conditions, uncertainties in external climate forcing (e.g., surface mass balance and ocean-induced melting) or uncertain input data such as the ice sheet geometry (e.g., bedrock topography and surface elevation) or boundary conditions (e.g., basal friction and geothermal heat flux). In order to provide more robust and reliable model-based estimates of ice-sheet mass balance, we therefore need to understand how model outputs are affected by or sensitive to input parameters.

To this aim, uncertainty quantification (UQ) methods have become a powerful and popular tool to deduce the impact of sources of uncertainty on ice-sheet projections (propagation of uncertainty) or to ascertain and rank the impact of each source of uncertainty on the projection uncertainty (sensitivity analysis) (Bulthuis et al., 2019, 2020; Edwards et al., 2019, 2021; Larour et al., 2012b; Ritz et al., 2015; Schlegel et al., 2018). While the aforementioned studies have mainly focused on the propagation and sensitivity analysis of sources of uncertainty in ice-sheet models, relatively little attention has been paid to the preliminary and critical characterization and description of these sources of uncertainty. Uncertainties in input parameters are generally lumped into uncertain (lumped) parameters that represent one or several sources of parametric and/or unmodeled dynamics uncertainty (Bulthuis et al., 2019; Edwards et al., 2019, 2021; Ritz et al., 2015). These uncertain lumped parameters then appear in parameterizations and reduced-order models of complex physical processes such as basal friction or ocean-induced melting. However, such a characterization does not reflect the fact that most sources of uncertainty in numerical ice-sheet models generally vary in space and potentially in time. With the aim of sampling spatially distributed input variables, Larour et al. (2012b) implemented an approach based on a partitioning of the computational domain, with later applications in Larour et al. (2012a), Schlegel et al. (2013, 2015, 2018), and Schlegel and Larour (2019). In this approach the computational domain is divided into a number of partitions fixed by the user or randomly determined based on the CHACO mesh partitioner (Hendrickson and Leland, 1995). For each partition, a statistical distribution is specified for the field being sampled; thus, sampling is not carried out over the entire field but over the multiple partitions. The number of partitions controls the spatial scale of the random perturbation; the larger the number of the partitions, the more local the perturbation.

In this paper, we propose to characterize uncertain spatially varying input parameters as spatial random fields, that is, an infinite set of random variables indexed by the spatial coordinate. More specifically, we focus on the class of Gaussian random fields, which provides a popular statistical model to represent stochastic phenomena in engineering, spatial analysis, and geostatistics (e.g., kriging interpolation) (Cressie, 1993; Schabenberger and Gotway, 2004). Gaussian random fields are entirely defined by their mean, or trend function, and their covariance function. The mean function is typically used to capture the dependance of a stochastic phenomenon at large scales while the covariance function specifies the spatial dependency between pairs of locations. Among the families of covariance functions, we focus on the Matérn family which provides a popular and practical choice to represent spatial correlations in geostatistics due to its high flexibility (Stein, 1999). In particular,





the Matérn family involves a set of parameters that control the correlation length and smoothness of the realizations of the random field.

Typical methods for sampling from Gaussian random fields, for instance methods based on a factorization of the covariance matrix, can be computationally intensive for large-scale problems or not appropriate for unstructured meshes, as can be en-
countered in applications in glaciology. Here, we implement an efficient sampling approach based on an explicit link between Gaussian random fields with Matérn covariance function and a stochastic partial differential equation (SPDE) (Lindgren et al., 2011). Discretization of this SPDE with the finite element method (FEM) results in a sparse and computationally efficient representation of a Gaussian random field known as a Gaussian Markov random field (GRMF). In addition, spatio-temporal samples can be generated by combining a temporal autoregressive model and the Matérn random field (Cameletti et al., 2012).
We carry out the implementation and analysis within the Ice-sheet and Sea-level System Model (ISSM) (Larour et al., 2012c) but our approach can be applied to other finite-element ice-sheet models.

This paper is organized as follows. In Sect. 2, we review the link between Gaussian random fields with Matérn covariance function and a stochastic partial differential equation (SPDE) and describe its discretization with the FEM. In Sect. 3, we test the implementation of the SPDE on a set of synthetic experiments to verify that it captures well the desired spatial and temporal
correlations. In Sect. 4, we provide an illustration concerned with assessing the impact of various sources of uncertainties on the Pine Island Glacier (PIG), West Antarctica. Finally, we provide an overall discussion of the results and the approach in Sect. 5.

## 2   Theory and Methods

In this work, we model a spatially varying uncertain input parameter as a random field $\{x(\boldsymbol{s}), \boldsymbol{s} \in D\}$ indexed over the com-
putational domain $D$. We decompose this random field as

$$x(\boldsymbol{s}) = \mu(\boldsymbol{s}) + \epsilon(\boldsymbol{s}), \tag{1}$$

where $\mu$ is the (deterministic) mean function and $\epsilon$ represents a stochastic term with zero mean. In the following, we assume without loss of generality the mean function to be zero, so that $x(\boldsymbol{s}) = \epsilon(\boldsymbol{s})$, and we model the stochastic term as a Gaussian random field.

### 2.1   Matérn covariance function

A Gaussian random field $\{x(\boldsymbol{s}), \boldsymbol{s} \in D\}$ with zero mean is totally characterized by its covariance or kernel function $C(\boldsymbol{s}, \boldsymbol{s}')$, which is defined mathematically by $\mathbb{E}[x(\boldsymbol{s})x(\boldsymbol{s}')]$, where $\mathbb{E}$ denotes the mathematical expectation. The covariance function describes the statistical relationship (or spatial correlation) between two locations $\boldsymbol{s}$ and $\boldsymbol{s}'$ in the domain. Roughly speaking, it describes the similarity of the values taken by the random field at different locations in the domain, with locations close to each other more likely to have similar values. In theory, any positive semidefinite function $C(\boldsymbol{s}, \boldsymbol{s}')$ defines a valid covariance





function (Rasmussen and Williams, 2006). In practice, the covariance function is often assumed to be a function of the space lag $\boldsymbol{s}' - \boldsymbol{s}$ (stationary covariance function) or even the spatial distance $\|\boldsymbol{s}' - \boldsymbol{s}\|$ (isotropic covariance function).

Among the families of covariance functions, the Matérn family is a popular choice to represent spatial correlation in geostatistics. Applications include spatial modeling of greenhouse gas emissions (Western et al., 2020), temperature or precipitation anomalies (Furrer et al., 2006), soil properties (Minasny and McBratney, 2005), acoustic wave speeds in seismology (Bui-Thanh et al., 2013), and basal friction in glaciology (Isaac et al., 2015; Petra et al., 2014). The Matérn covariance function is defined as

$$C(\boldsymbol{s}, \boldsymbol{s}') = \frac{\sigma^2}{2^{\nu-1}\Gamma(\nu)} (\kappa\|\boldsymbol{s} - \boldsymbol{s}'\|)^{\nu} K_{\nu}(\kappa\|\boldsymbol{s} - \boldsymbol{s}'\|), \tag{2}$$

where $K_{\nu}$ is the modified Bessel function of the second kind (Abramowitz and Stegun, 1970) and order $\nu > 0$, $\Gamma$ the Gamma function, $\sigma^2$ the variance of the random field, and $\kappa$ a positive scaling parameter. The scaling parameter $\kappa$ is more naturally interpreted as a range parameter $\rho$, with the empirically derived relationship $\rho = \sqrt{8\nu}/\kappa$ corresponding to correlations near $0.1$ at the distance $\rho$ (Lindgren et al., 2011). The order $\nu$, or smoothness index, controls the smoothness of the realizations; more specifically, the realizations are $\lceil \nu - 1 \rceil$ times differentiable. Therefore, the order parameter $\nu$ allows for additional flexibility in spatial modeling, giving the Matérn family considerable practical value (Stein, 1999). As $\nu \to \infty$, the Matérn covariance function approaches the squared exponential (also called the Gaussian) form, popular in machine learning, whose realizations are infinitely differentiable. For $\nu = 0.5$, the Matérn covariance function takes the exponential form, popular in spatial statistics, which produces continuous but non-differentiable realizations.

### 2.1.1 Stochastic partial differential equation

In this paper, interest lies in generating realizations of Gaussian random fields for uncertainty quantification in ice-sheet models. Various methods exist to sample Gaussian random fields, see for instance Hristopulos (2020); yet these methods can be computationally prohibitive for applications in glaciology. For instance, methods based on a factorization of the covariance matrix typically results in large dense matrices when the number of degrees of freedom is large. Here, we consider a sampling approach that is based on an explicit link between Gaussian random fields with Matérn covariance function and a stochastic partial differential equation. Indeed, as noted initially in Whittle (1954) and more recently revisited in Lindgren et al. (2011), realizations of a Gaussian random field $x(\mathbf{s})$ with Matérn covariance function can be obtained as the solution to the fractional stochastic partial differential equation

$$(\kappa^2 - \Delta)^{\alpha/2}(\tau x(\boldsymbol{s})) = \mathcal{W}(\boldsymbol{s}), \tag{3}$$

where $\kappa$ and $\tau$ are positive parameters, $\alpha = \nu + d/2$ (where $d$ is the spatial dimension), and $\mathcal{W}(\mathbf{s})$ is a spatial Gaussian white noise with unit variance. The parameter $\kappa$ controls the range parameter $\rho$ and the parameters $\kappa$ and $\tau$ control the variance of





the random field. More explicitly, we have the following relationships (see, for instance, Khristenko et al. (2019)):

$$\nu = \alpha - \frac{d}{2}, \tag{4a}$$

$$\rho = \frac{\sqrt{8\nu}}{\kappa}, \tag{4b}$$

$$\sigma^2 = \frac{\Gamma(\nu)}{\Gamma(\alpha)(4\pi)^{d/2}\kappa^{2\nu}\tau^2}. \tag{4c}$$

An intuitive interpretation for the rationale behind the SPDE (3) can be found in the context of the filter theory. For $\alpha = 2$, the differential operator $(\kappa^2 - \Delta)$ can be interpreted as a low-pass filter or diffusion operator that smooths out a Gaussian white noise (no spatial correlation) into a Gaussian random field that exhibits spatial correlation. While a Gaussian white noise is characterized by a flat wavenumber spectrum, a filtered random field exhibits a decaying wavenumber spectrum, with $\kappa^2$ acting as a cutoff parameter.

Equation (3) requires a choice of boundary conditions. Here, we consider either homogeneous Neumann boundary conditions

$$\nabla x(\boldsymbol{s}) \cdot \boldsymbol{n} = 0, \tag{5}$$

or Robin boundary conditions

$$\beta x(\boldsymbol{s}) + \nabla x(\boldsymbol{s}) \cdot \boldsymbol{n} = 0, \tag{6}$$

where $\beta$ is the Robin coefficient and $\boldsymbol{n}$ is the external unit vector to the boundary. We mention that the choice of boundary conditions typically introduces boundary effects that lead to increased or decreased correlation and pointwise variance close to the boundary; see, for instance, Daon et al. (2018) and Khristenko et al. (2019) for a discusion and approaches to reduce the impact of boundary effects. For $\alpha = 2$ and Robin boundary conditions, the choice of $\beta = (\kappa\tau)/1.42$ has been found to be a good choice to mitigate boundary effects (Roininen et al., 2014).

The SPDE approach has several advantages compared to other sampling methods. First, Eq. (3) can be solved efficiently by using standard discretization schemes (e.g, the finite element method) and sparse matrix linear algebra (see Sect. 2.1.2). Second, the SPDE approach provides a flexible model to build more general probabilistic models of random fields that do not require to impose the covariance structure explicitly; see Simpson et al. (2011) for a discussion. For instance, the parameters $\kappa$ and $\tau$ in Eq. (3) can vary spatially in order to represent non-stationary Gaussian random fields with spatially varying range parameter and marginal variance. Similarly, the Robin coefficient can vary spatially to mitigate boundary effects (Daon et al., 2018). Other extensions of the SPDE approach include anisotropic Gaussian models (Lindgren et al., 2011) or even non-Gaussian models (Bolin, 2014). Finally, the SPDE approach can be extended to define Gaussian random fields with Matérn covariance function on general spatial domains such as the sphere (Lindgren et al., 2011).





### 2.1.2 Numerical implementation

We assume at this stage that $\alpha = 2$. The SPDE (3) can be solved efficiently by the finite element method; see, for instance, Lindgren et al. (2011) and Chu and Guilleminot (2019). The finite element representation of the solution writes as

$$x(\boldsymbol{s}) = \sum_{k=1}^{N} x_k \psi_k(\boldsymbol{s}), \tag{7}$$

where $\{\psi_k\}_{k=1}^N$ is a finite element basis (here piecewise linear functions), $N$ the total number of nodes, and $\{x_k\}_{k=1}^N$ are the stochastic nodal values. The discretized weak form of the differential operator in Eq. (3) with Robin boundary conditions

involves the matrix $\mathbf{K}$ with entries

$$K_{ij} = \int_D \nabla \psi_i(\boldsymbol{s}) \cdot \nabla \psi_j(\boldsymbol{s}) d\boldsymbol{s} + \int_D \kappa^2(\boldsymbol{s}) \psi_i(\boldsymbol{s}) \psi_j(\boldsymbol{s}) d\boldsymbol{s} + \int_{\partial D} \beta(\boldsymbol{s}) \psi_i(\boldsymbol{s}) \psi_j(\boldsymbol{s}) d\boldsymbol{s}, \tag{8}$$

where the test functions are the same as the basis functions and $\partial D$ denotes the boundary of the domain $D$. The finite element discretization of the SPDE also involves the mass matrix $\mathbf{M}$, whose entries are given by

$$M_{ij} = \int_D \psi_i(\boldsymbol{s}) \psi_j(\boldsymbol{s}) d\boldsymbol{s}. \tag{9}$$

As shown in Lindgren et al. (2011), the vector $\boldsymbol{x}$ of nodal values of the FEM solution to the SPDE (3) is a Gaussian random vector that satisfies

$$\boldsymbol{x} \sim \mathcal{N}(\boldsymbol{0}, \boldsymbol{\Sigma}^{(2)}), \tag{10}$$

where the covariance matrix is given by

$$\boldsymbol{\Sigma}^{(2)} = \mathbf{K}^{-1} \mathbf{M} \mathbf{K}^{-1}, \tag{11}$$

where the superscript underpins the order 2. Let $\boldsymbol{\Sigma}^{(2)} = \mathbf{L}^{(2)} \mathbf{L}^{(2)^T}$ be a matrix factorization of $\boldsymbol{\Sigma}^{(2)}$ (e.g., Cholesky or QR decomposition). Then, realizations of $\boldsymbol{x}$ are obtained as

$$\boldsymbol{x} = \mathbf{L}^{(2)} \boldsymbol{\xi}, \tag{12}$$

where $\boldsymbol{\xi}$ is a random vector whose entries are independent standard Gaussian random variables. Equivalently, one can show that realizations of $\boldsymbol{x}$ can obtained by solving the following linear system of equations

$$\mathbf{K}\boldsymbol{x} = \mathbf{M}^{1/2} \boldsymbol{\xi}. \tag{13}$$

More generally, the covariance matrix $\boldsymbol{\Sigma}^{(\alpha)}$ for any arbitrary integer order $\alpha$ is given by (Lindgren et al., 2011)

$$\boldsymbol{\Sigma}^{(\alpha)} = \mathbf{K}^{-1} \mathbf{M} \boldsymbol{\Sigma}^{(\alpha-2)} \mathbf{K}^{-1} \mathbf{M}, \tag{14}$$





where $\boldsymbol{\Sigma}^{(1)} = \mathbf{K}^{-1}$ and $\boldsymbol{\Sigma}^{(2)} = \mathbf{K}^{-1}\mathbf{M}\mathbf{K}^{-1}$. Also the matrix $\mathbf{L}^{(\alpha)}$ for any arbitrary integer order $\alpha$ is given by

$$\mathbf{L}^{(\alpha)} = \mathbf{K}^{-1}\mathbf{M}\mathbf{L}^{(\alpha-2)}, \tag{15}$$

where $\mathbf{L}^{(1)} = \mathbf{K}^{1/2}$ and $\mathbf{L}^{(2)} = \mathbf{K}^{-1}\mathbf{M}^{1/2}$. Therefore, samples for any arbitrary integer order $\alpha > 2$ can be generated in a recursive way. The algorithm for sampling Gaussian random fields with Matérn covariance function based on the SPDE approach is summarized in Algorithm 1. Although not implemented in our work, we mention the work by Bolin et al. (2018) that extends the numerical solution of the SPDE (3) to non-integer orders.

---

**Algorithm 1** SPDE-based sampling algorithm for Gaussian random fields with Matérn covariance function

---

Set input data $\alpha$, $\tau$, $\kappa$ and $\beta$;

Create FEM discretization: Build stiffness matrix $\mathbf{K}$ and mass matrix $\mathbf{M}$;

Create random vector $\boldsymbol{\xi}$ with independent entries $\sim \mathcal{N}(0,1)$;

**if** $\alpha\%2 \neq 0$ **then**

  Compute the square root of $\mathbf{K}$;

  Solve $\mathbf{K}\boldsymbol{x}^{(1)} = \mathbf{K}^{1/2}\boldsymbol{\xi}$;

**else**

  Compute the square root of $\mathbf{M}$

  Solve $\mathbf{K}\boldsymbol{x}^{(2)} = \mathbf{M}^{1/2}\boldsymbol{\xi}$;

**end if**

**for** $i = 3$; $i \leq \alpha$; $i = i + 2$ **do**

  Solve $\mathbf{K}\boldsymbol{x}^{(i)} = \mathbf{M}\boldsymbol{x}^{(i-2)}$;

**end for**

Set $\boldsymbol{x}^{(\alpha)} \leftarrow \boldsymbol{x}^{(\alpha)}/\tau$;

**return** Sample $\boldsymbol{x}^{(\alpha)}$;

---

In order to speed up the computation, the mass matrix $\mathbf{M}$ can be approximated as a diagonal lump mass matrix $\widetilde{\mathbf{M}}$ with

diagonal entries

$$\widetilde{M}_{ii} = \sum_{k=1}^{N} M_{ij}, 1 \leq k \leq N. \tag{16}$$

Numerically, this approximation leads to a representation of the precision matrix, i.e., the inverse of the covariance matrix, that is sparse (Lindgren et al., 2011); elements in the precision matrix are non zero only for diagonal elements and neighboring nodes. Mathematically, such a sparse representation of the precision matrix corresponds to a Gaussian Markov random field

(GRMF) (Rue and Held, 2005). This means that the conditional expectation of the random field at a given node given all the other nodes only depends on the neighboring nodes.





### 2.1.3 Extension to spatio-temporal random fields

Temporal correlation between samples (for transient simulation) is represented with a first-order autoregressive model (AR1 process) (Cameletti et al., 2012; Western et al., 2020):

$$x_t(\boldsymbol{s}) = \phi x_{t-1}(\boldsymbol{s}) + \epsilon_t(\boldsymbol{s}), \tag{17}$$

where $t$ denotes a discrete time, $\phi$ the temporal correlation (can be positive, negative, or null), and $\epsilon_t(\boldsymbol{s})$ an independent noise term. The first-order autoregressive model specifies that the value of the random field at time $t$ depends linearly on its value at time $t-1$ and a stochastic term that models unpredictable effects. The parameter $\phi$ controls the strength (with sign) of the temporal correlation between the values of the random field at times $t$ and $t-1$. If $\phi = 0$, there is no dependence between

$x_t$ and $x_{t-1}$ and only the stochastic term contributes to the spatio-temporal random field. By definition, an AR1 model is an example of a Markov process.

At every time step $t$, the noise term $\epsilon_t(\boldsymbol{s})$ is chosen as a Gaussian random field with Matérn covariance function, obtained as the solution of the SPDE (3). If $x_0$ is a stationary Gaussian random field with variance $\sigma^2$ and $\epsilon_t$ is a stationary Gaussian random field with variance $\sigma_\epsilon^2 = \sigma^2(1 - \phi^2)$, then $x_t$ is a stationary process in time with zero mean and marginal variance $\sigma^2$

if $|\phi| < 1$. The process is non stationary if $|\phi| \geq 1$, with the case $\phi = 1$ corresponding to a discrete random walk. The temporal correlation function, or autocorrelation function, is given by

$$\mathbb{E}[x_t x_{t+T}] = \frac{\sigma_\epsilon^2}{1 - \phi^2} \phi^{|T|}, \tag{18}$$

and is therefore stationary because it only depends on the time lag $T$. Algorithm (2) provides a pseudo-code to generate spatio-temporal samples.

The spatio-temporal model combining Eqs. (3) and (17) defines a spatio-temporal Gaussian random field discretized in time. It represents an example of a separable model in which the spatio-temporal covariance function is the product of the spatial and temporal covariance functions. Therefore, the spatial and temporal covariance structures of the model are totally decoupled; the temporal evolution of the input quantity at a given location does not depend on the temporal evolution at other locations. Although beyond the scope of this work, non-separable models can however be formulated based on the SPDE approach

(Krainski, 2018; Bakka et al., 2020).

## 3 Sampling capability: Verification

As a means of verifying and testing the implementation of our new sampling capability, we carry out a set of numerical experiments on a synthetic ice sheet/ice shelf of size $100\,\text{km} \times 100\,\text{km}$. In these experiments, we consider Matérn random fields with a parameter range of $20\,\text{km}$ and a unit variance. We first verify the spatial covariance structure by solving the

SPDE (3) for the orders $\alpha = 2$ and $4$, then discuss the impact of the boundary conditions on the simulations in a second experiment. We finally perform transient simulations to verify the temporal covariance structure.





---

**Algorithm 2** AR1 algorithm for transient Gaussian random fields with Matérn covariance function

---

Set input data $\alpha$, $\tau$, $\kappa$ and $\beta$ for the stochastic term $\epsilon_t$, $\phi$ for the temporal correlation, and time step and final time for transient;

Create FEM discretization: Build stiffness matrix $\mathbf{K}$ and mass matrix $\mathbf{M}$;

**Initialization**: Set time $= 0$, $t = 0$, and initial sample $\boldsymbol{x}_0$;

**while** time $<$ finaltime **do**

    time = time + timestep, t=t+1;

    Generate $\boldsymbol{\epsilon}_t$ using Algorithm 1;

    Set $\boldsymbol{x}_t = \phi \boldsymbol{x}_{t-1} + \boldsymbol{\epsilon}_t$;

**end while**

**return** Sample $\boldsymbol{x}_1, \boldsymbol{x}_2, \ldots$.

---

### 3.1 Experiment 1: Verification of covariance structure

As a first experiment, we test our implementation by checking that the generated samples are actually drawn from a Gaussian random field with the desired Matérn covariance function. More specifically, the marginal distributions have to follow
Gaussian distributions with zero mean and unit variance while the bivariate distributions have to follow bivariate Gaussian distributions with zero mean and covariance given by Eq. (2). To test for the covariance structure, we generate an ensemble of 10 000 independent and identically distributed (i.i.d) samples and estimate the marginal and bivariate distributions from these samples. Figures 1 and 2 show the obtained distributions for the locations $\boldsymbol{s}_1 = (50, 50)$ km, $\boldsymbol{s}_2 = (53.125, 50)$ km, and $\boldsymbol{s}_3 = (59.375, 50)$ km for $\alpha = 2$ and $\alpha = 4$, respectively. In both figures, we observe that the marginal and bivariate distributions
all follow a Gaussian distribution with variances and covariances close to the theoretical expected values. Small discrepancies between the estimated variances and covariances and their theoretical values can be explained by the numerical approximations (finite element discretization and lumped approximations), the impact of boundary conditions (discussed in Experiment 2), and the limited number of samples used for the estimation.

### 3.2 Experiment 2: Influence of the boundary conditions

The second experiment investigates the influence of the boundary conditions on the marginal variance of the samples generated following the SPDE approach. Figure 3 shows the estimated marginal standard deviation for $\alpha = 2$ with 10000 samples considering either homogeneous Neumann boundary conditions (a) or Robin boundary conditions (b). The value of the Robin coefficient $\beta$ is taken as $\kappa\tau/1.42$. With homogeneous Neumann boundary conditions, we observe a clear boundary artifact that yields a larger marginal standard deviation close to the boundary than in the interior. In particular, the standard deviation
is 1.4 times larger along the edges of the domain and 2 times larger near the corners. The boundary artifact becomes almost negligible near a distance of at least $\rho$ from the boundary. On the contrary, Robin boundary conditions with an appropriate choice of the Robin coefficient help mitigate the boundary artifact significantly. Physically, Neumann boundary conditions can



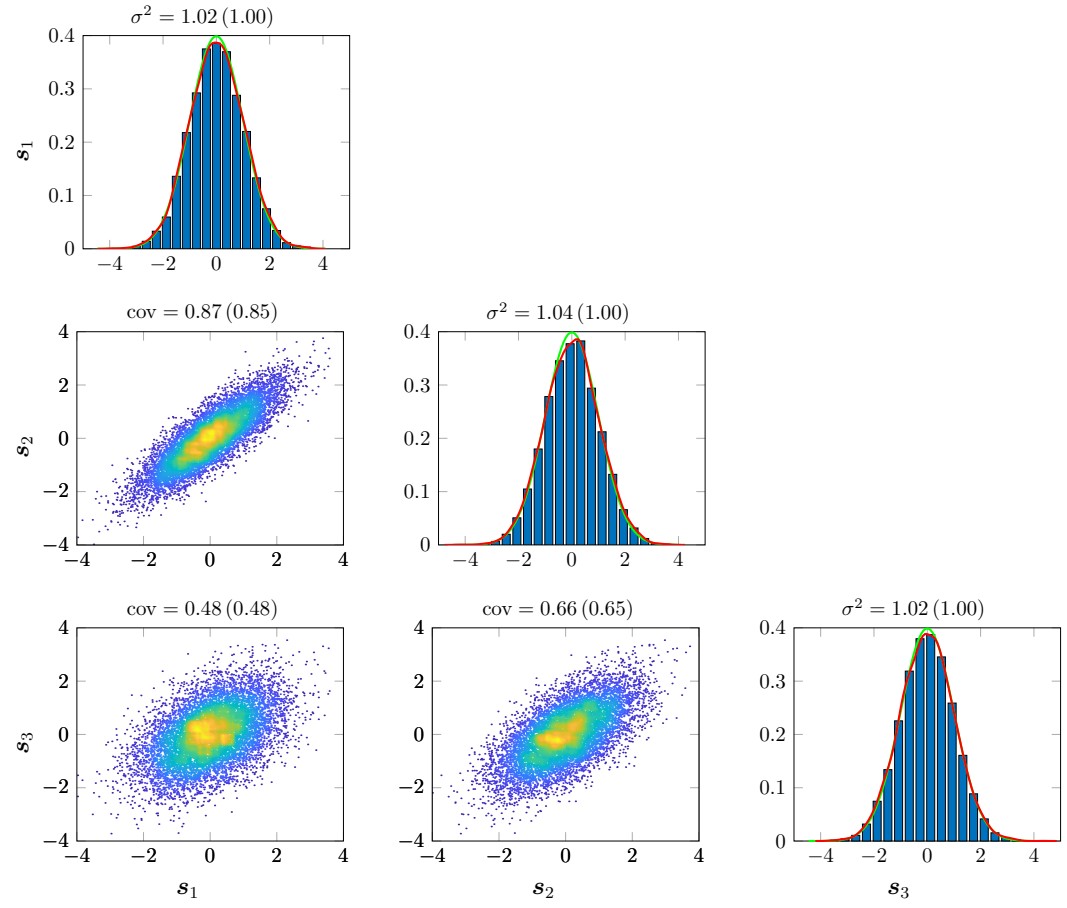

**Figure 1.** Estimated marginal and bivariate distributions of the samples generated by solving the SPDE (3) with $\alpha = 2$. Estimates are based on 10000 samples. Results are shown for the locations $\boldsymbol{s}_1 = (50, 50)$ km, $\boldsymbol{s}_2 = (53.125, 50)$ km, and $\boldsymbol{s}_3 = (59.375, 50)$ km. The estimated covariances and variances are also provided with the theoretical values determined from Eq. (2) between brackets. For the marginal distributions, the red curves represent the kernel density estimates and the green curves are standard Gaussian distributions.

be interpreted as reflecting boundaries, while Robin boundary conditions can be understood as absorbing boundaries, with the Robin coefficient being an absorption coefficient.

## 3.3 Experiment 3: Transient simulation

As a third experiment, we test our transient implementation by checking that the generated samples follow an AR1 process in time. For different values of the temporal correlation factor $\phi$, we generate a series of 10000 spatio-temporal samples and estimate the autocorrelation function at the center location $\boldsymbol{s} = (50, 50)$ km. Figure 4 represents the estimated autocorrelation functions and provides a comparison with the theoretical autocorrelation functions given by Eq. (18). For all temporal corre-



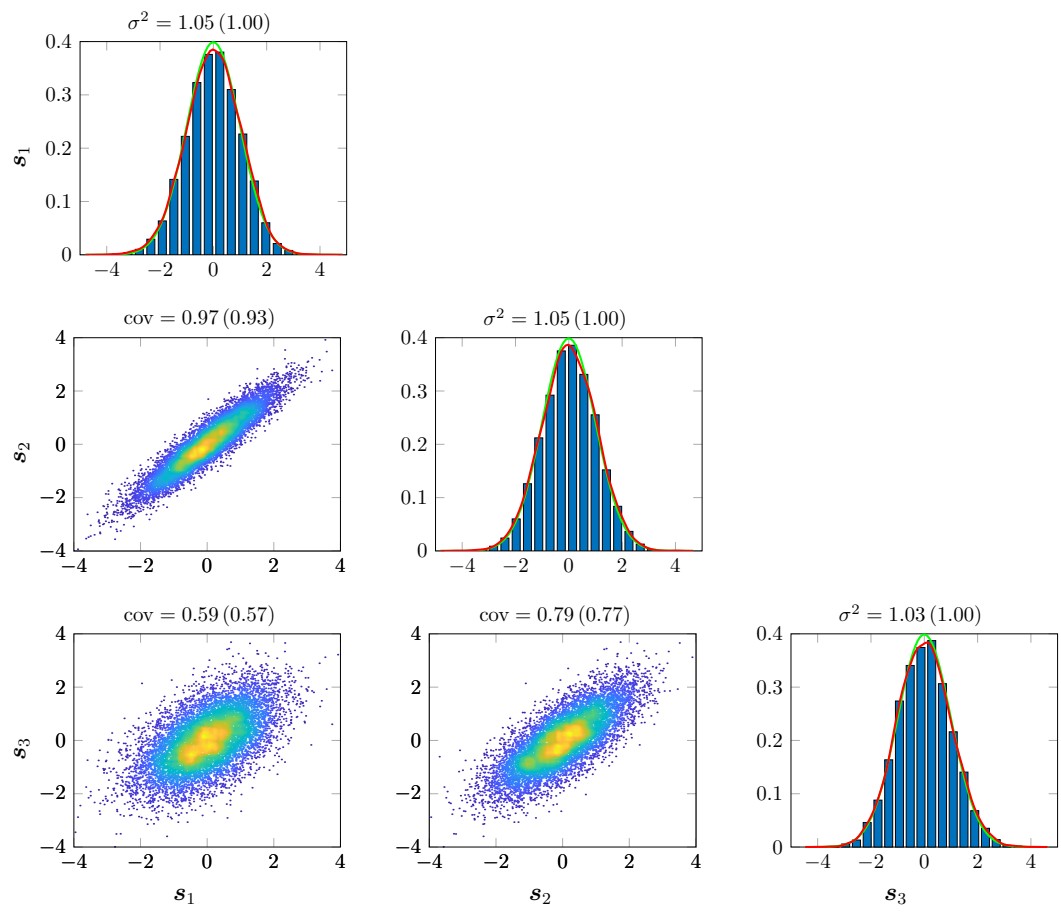

**Figure 2.** Same as Fig. 1 but for $\alpha = 4$.

lation factors, the estimated autocorrelation functions are close to their theoretical value. The small discrepancies can also be explained by numerical approximations and the limited number of samples in the time series.

## 4 Sampling capability: Application

### 4.1 Experimental setup

We set up a test problem similar to the application in Larour et al. (2012b), in which the authors performed a UQ analysis on the Pine Island Glacier using a partitioning approach. We refer the reader to this article for additional information about the ice flow model, the datasets, the input quantities, and the quantities of interest. We consider three spatially varying uncertain input quantities: the ice thickness $H$, the basal drag coefficient $\alpha$ at the ice/bed interface, and the ice hardness $B$. The basal drag coefficient is used in a Budd-type friction law and the ice hardness is determined from a thermal model (Larour et al., 2012c).



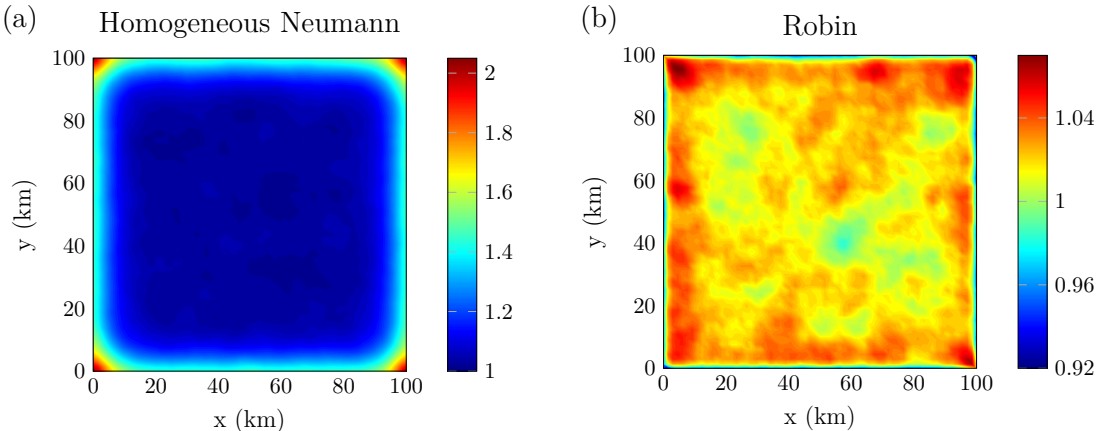

**Figure 3.** Comparison of estimated marginal standard deviations with homogeneous Neumann boundary conditions (a) and Robin boundary conditions (b) with Robin coefficient given by $\beta = (\kappa\tau)/1.42$.

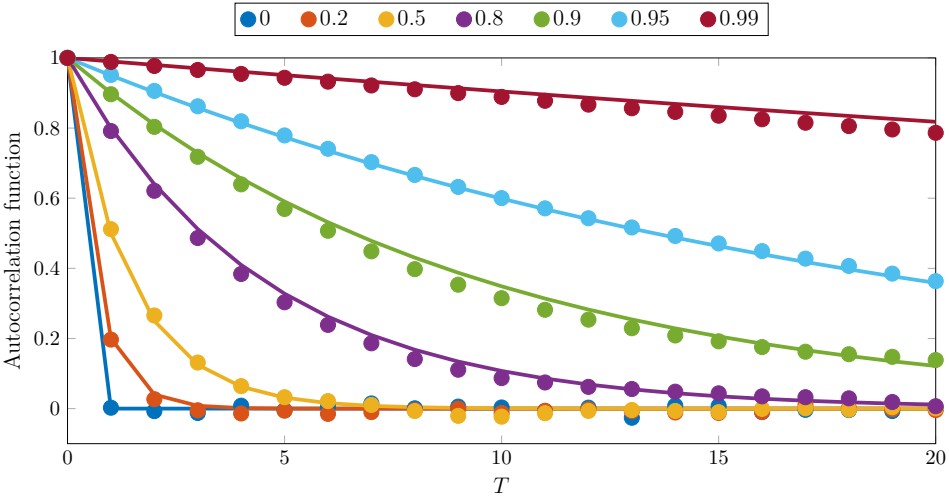

**Figure 4.** Estimated autocorrelation functions (dot lines) for different values of the temporal correlation factor $\phi$ as a function of the temporal lag $T$. Solid lines represent the theoretical autocorrelation functions given by Eq. (18).

As quantities of interest, we consider the mass flux through thirteen fluxgates positioned along the main tributaries of the PIG
(see Figure 5). The mass flux $M_i$ through gate $i$ is given by

$$M_i = \int_{\mathcal{C}_i} \rho_{\text{ice}} H \boldsymbol{v} \cdot \boldsymbol{n} \, dl, \tag{19}$$

where the integral is a line integral along the fluxgate contour $\mathcal{C}_i$, $\boldsymbol{v}$ the depth-averaged horizontal velocity computed using the
Shallow Shelf-Stream Approximation (SSA) (MacAyeal, 1989), $\boldsymbol{n}$ the downstream unit normal vector to the fluxgate contour,





and $\rho_{\mathrm{ice}}$ the ice density. All thirteen mass fluxes are computed simultaneously for each forward run of ISSM and each sample
of the input quantities.

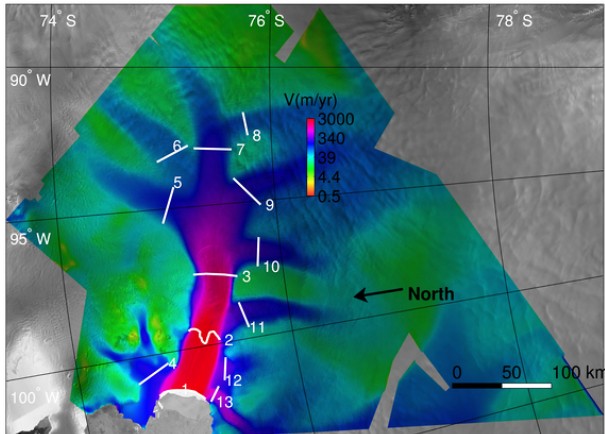

**Figure 5.** Fluxgates used to compute mass fluxes on tributaries of the PIG. Each gate is numbered from 1 to 13, and corresponds to one tributary. Gate 1 coincides with the ice front, and gate 2 coincides with the 1996 grounding line. The gates are superimposed on an InSAR surface velocity map of the area, in logarithmic scale [Rignot, 2008]. From Larour et al. (2012b), with courtesy of Journal of Geophysical Research: Earth Surface.

## 4.2  Characterization of uncertainty

We represent each source of uncertainty in the following way

$$q(\boldsymbol{s}) = q_{\mathrm{ref}}(\boldsymbol{s})(1 + \sigma_q(\boldsymbol{s})p_q(\boldsymbol{s})), \tag{20}$$

where $q_{\mathrm{ref}}$ is a reference value for each source of uncertainty, $\sigma_q$ the (spatially distributed) error margin for each source of
uncertainty, and $p_q(\boldsymbol{s})$ is a spatially distributed random perturbation. The reference thickness value is derived from bedrock topography (Nitsche et al., 2007) and surface elevation (Bamber et al., 2009; Griggs and Bamber, 2009) data. The temperature-dependent reference ice hardness value is derived from an Arrhenius law (Larour et al., 2012c) and surface temperature data by Comiso (2000). The reference basal drag coefficient is inferred using an inverse method (Morlighem et al., 2010) with the reference thickness and ice hardness values. For the UQ analysis in Sect. 4.3 and 4.4, we only perturb the ice thickness and
the error margins are based on Ground Penetrating Radar (GPR) cross-over measurements from the 2009 Operation IceBridge Campaign and provided by the Center for Remote Sensing of Ice Sheet (CReSIS); see also Larour et al. (2012b) for further information. For the sensitivity analysis in Sect. 4.5, we specify a spatially uniform 5% error margin on all three inputs to compare sensitivities between input variables with same error margins.

Here, we represent the random perturbation $p_q$ as a Gaussian random field with Matérn covariance function and unit variance
($\sigma^2 = 1$) following the SPDE approach. Therefore, each uncertain input quantity is represented as a Gaussian random field with

mean $q_{\mathrm{ref}}$, Matérn covariance function and marginal standard deviation proportional to $q_{\mathrm{ref}}(\boldsymbol{s})\sigma_q(\boldsymbol{s})$. We solve the SPDE (3) for $\alpha = 2$ and impose Robin boundary conditions with $\beta = (\kappa\tau)/1.42$. We consider different parameter ranges $\rho$ to investigate the impact of the spatial correlation on the results. The parameters $\kappa$ and $\tau$ are determined from Eqs. (4a)–(4c). As an illustration, we represent samples of the random perturbation for $\rho = 10$ km in Fig. 6, $\rho = 30$ km in Fig. 7, and $\rho = 100$ km in Fig. 8.

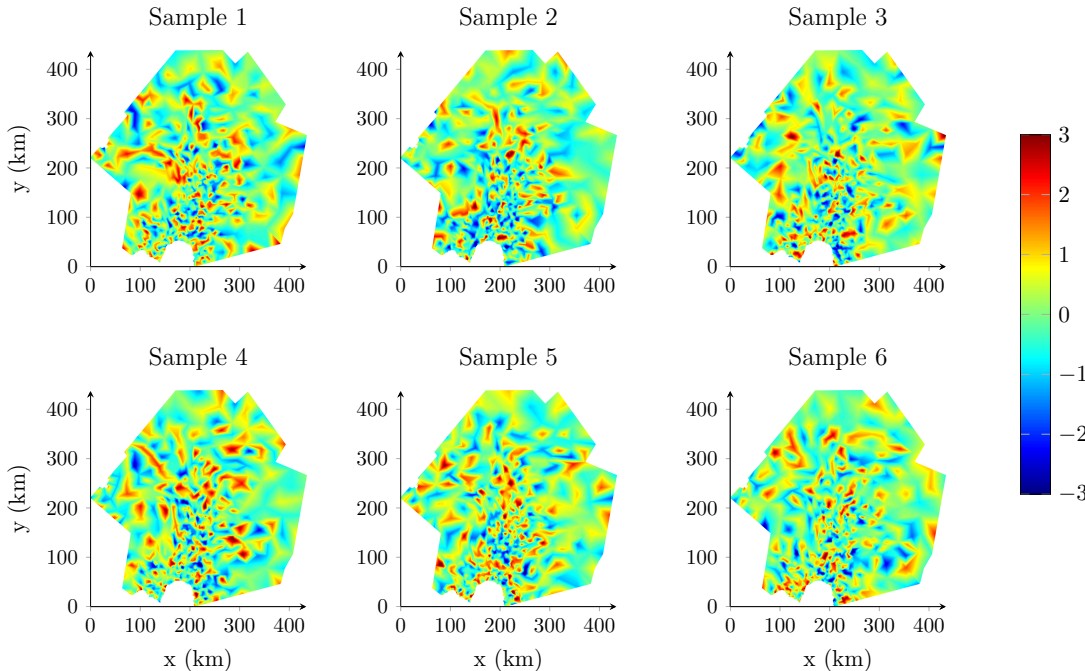

**Figure 6.** Samples of the random perturbation $p_q(\boldsymbol{s})$ over Pine Island Glacier with a parameter range of 10 km.

## 4.3 Propagation of uncertainty

We use ISSM to propagate uncertainty in ice thickness into a probabilistic characterization of the depth-averaged mass flux across fluxgates. We implement the propagation of uncertainty using Monte Carlo sampling. To this end, we begin by generating an ensemble of $n$ i.i.d samples of the random perturbation by solving the stochastic equation (3) for an ensemble of $n$ i.i.d samples of the random right-hand side $\mathcal{W}(\boldsymbol{s})$. Then, we run the ISSM computational model for each sample of the input variable to generate an ensemble of $n$ i.i.d samples of the depth-averaged mass flux across the 13 fluxgates. From these samples, we can estimate the mean and standard deviation of the mass flux across the fluxgates with the statistical mean and standard deviation of the samples. In addition, we can estimate the probability density distribution of the mass flux across the fluxgates by using a normalized histogram of the samples or kernel density estimation (Scott, 2015). In all our simulations, we found that $n = 2500$ samples were sufficient to ensure reasonably converged statistical estimates.



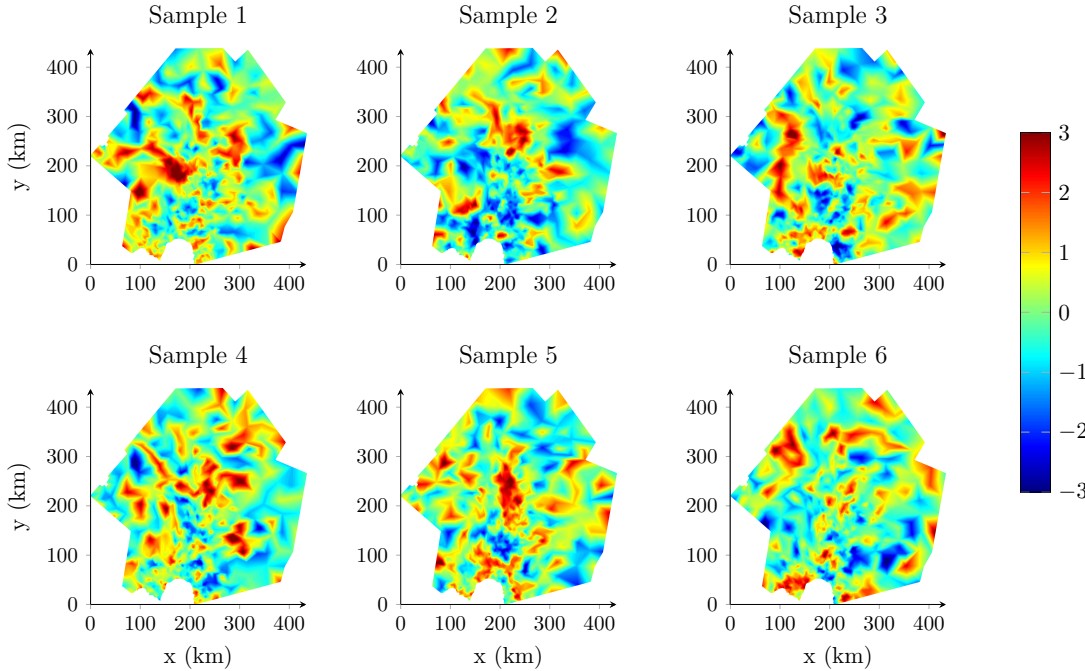

**Figure 7.** Same as Fig. 6 but with a parameter range of 30 km. For comparison, the SPDE (3) is solved with the same right-hand sides as in Fig. 6.

Figure 9 shows for a parameter range of 30 km the statistical distribution of mass flux through each gate as a normalized histogram of the samples along with a kernel density estimate of their probability density function. The estimated mean and coefficient of variation (defined as the standard deviation over the mean) are also provided. The statistical distribution of mass flux through each gate can be considered Gaussian (which can be checked formally by carrying out a Kolmogorov-Smirnov test), with the exception of gate 4, for which the distribution is negatively skewed. The estimated mean values are approximately

equal to the mass fluxes obtained for the reference ice thickness. The coefficients of variation range from of a few hundredths of percent to a few percent, with the fluxgates close to the ice front exhibiting the largest uncertainty. As already shown in Larour et al. (2012b), the mass flux response to input uncertainties is particularly smooth.

   Figure 10 shows the statistical distribution of mass flux through gate 3 for different parameter ranges $\rho$. We also present results for a white noise perturbation (zero correlation length) and a uniform Gaussian perturbation (infinite correlation length).

We observe that the mean value does not depend on the parameter range while the overall uncertainty (measured as the coefficient of variation) increases with the parameter range. With the exception of gate 4, similar conclusions can be drawn for all the other gates. The specific behavior of gate 4 may result from the non-Gaussianity of the mass flux distribution and the coefficient of variation is not the most appropriate measure of dispersion for skewed distributions.





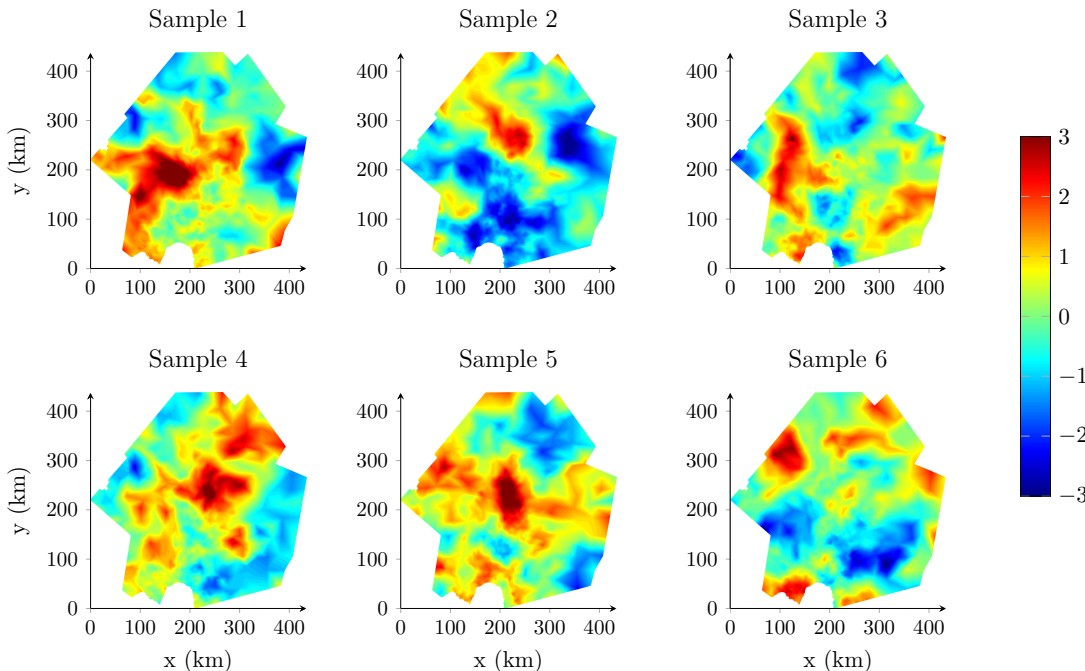

**Figure 8.** Same as Fig. 6 but with a parameter range of $100\,\mathrm{km}$. For comparison, the SPDE (3) is solved with the same right-hand sides as in Fig. 6.

### 4.4 Sensitivity analysis to ice thickness changes

We carry out a global sensitivity analysis to determine where changes in ice thickness most influence uncertainty in mass flux across fluxgates. For a given fluxgate, we can build a sensitivity map which gives for each location a sensitivity measure (or index) between the random field at this location and the mass flux across the fluxgate. Here, we consider as sensitivity measure the coefficient of correlation between the random field and the mass flux $M_i$, i.e.,

$$\frac{\mathrm{Cov}(p_q(\boldsymbol{s}), M_i)}{\sqrt{\mathbb{V}[p_q(\boldsymbol{s})]\mathbb{V}[M_i]}}, \tag{21}$$

where Cov denotes the covariance between two random variables and $\mathbb{V}$ the variance. We mention that other sensitivity measures like the Spearman's or Kendall rank correlations, the distance correlation sensitivity index, or the Hilbert-Schmidt independence criterion sensitivity index (Da Veiga, 2014; De Lozzo and Marrel, 2016) can also be considered as long as these indices can handle dependent variables.

Figure 11 shows for a parameter range of 30 km a sensitivity map for the mass flux through each gate. The coefficients of
correlation are estimated from the 2500 samples obtained in Sect. 4.3. A positive, respectively negative, index value indicates a positive, respectively negative, (linear) correlation between changes in ice thickness and changes in mass flux across a fluxgate, and a zero value no (linear) correlation. As expected, only a limited region on the glacier has a significant influence on the

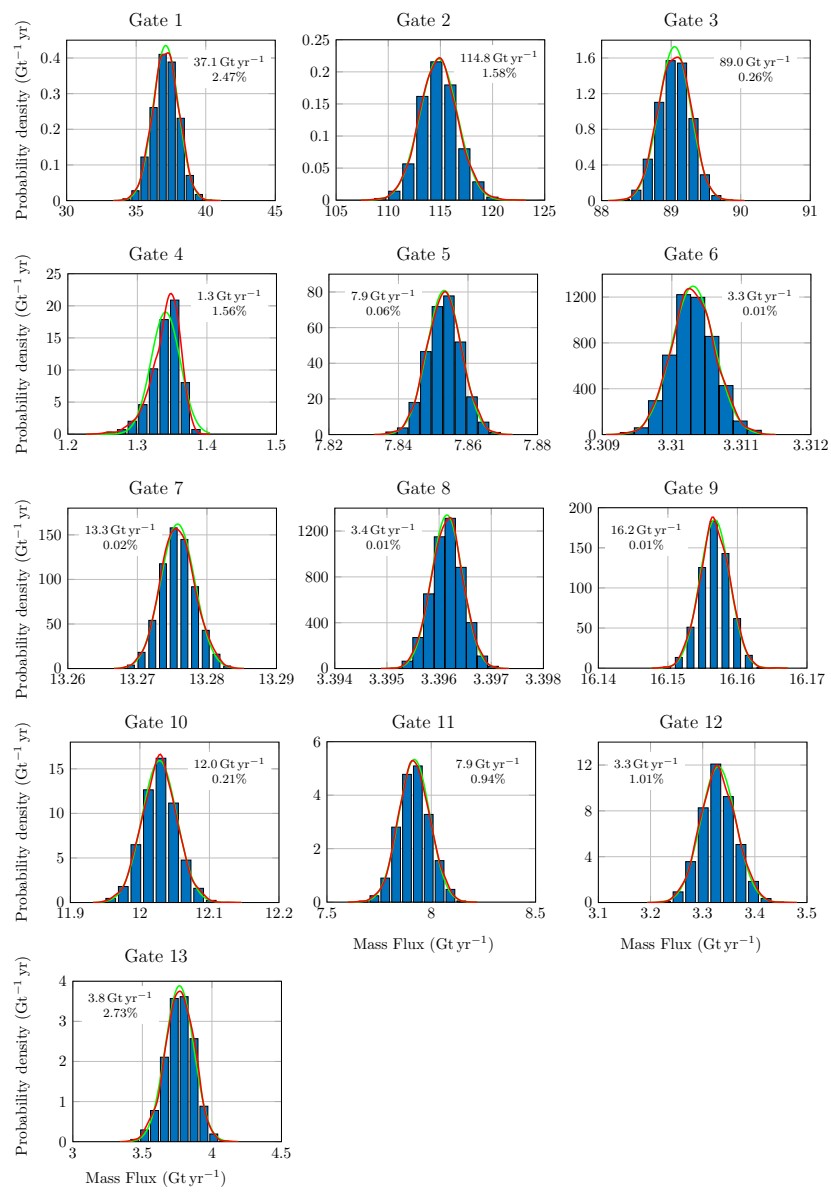

**Figure 9.** Normalized histograms for the mass flux across the fluxgates depicted in Figure 5. The mean (in $\mathrm{Gt\,yr^{-1}}$) and the coefficient of variation (in percent) are provided for each gate. The kernel density estimate of the probability density function is shown in red and its Gaussian approximation with corresponding mean and coefficient of variation in green. Results are obtained for a parameter range of 30 km.

mass flux across a given fluxgate. This region is usually concentrated close to the fluxgate though more remote locations can have an influence as can be seen for instance with fluxgates 6, 7, and 9. These remote locations lie in the main ice stream of the

PIG, thus suggesting that uncertainty in ice thickness in the main ice stream can impact ice flow in upstream tributaries. With





the exception of gate 4, sensitivity maps show a positive (linear) correlation between changes in ice thickness and changes in mass flux across fluxgates.

Figure 12 shows a sensitivity map of mass flux through gate 3 for different parameter ranges and white noise and uniform perturbations. As expected, increasing the parameter range expands the zone of influence and the magnitude of the correlation.
Therefore, the larger the correlation length, the further away from the gate uncertainties in ice thickness can influence mass flux estimates.

### 4.5   Sensitivity analysis to multiple input quantities

We perform another sensitivity analysis by considering multiple sources of uncertainty namely uncertainty in ice thickness, basal friction, and ice hardness. We assume that these three sources of uncertainty are statistically independent and we choose
a parameter range of 30 km for all of them. Because the sources of uncertainty are statistically independent, we can use the Sobol or variance-based sensitivity indices (Sobol', 2001; Saltelli et al., 2008) to characterize the contribution of each source of uncertainty to the uncertainty in the mass flux across each fluxgate. The first order Sobol index $S_{i,j}$ measuring the contribution of the random field $\{x_i(\boldsymbol{s}), \boldsymbol{s} \in D\}$ to the uncertainty in the mass flux $M_j$ is defined as

$$S_{i,j} = \frac{\mathbb{V}[\mathbb{E}[M_j | \{x_i(\boldsymbol{s}), \boldsymbol{s} \in D\}]]}{\mathbb{V}[M_j]}. \tag{22}$$

The Sobol index of a given uncertain input parameter represents the fraction of the variance of the projections explained as stemming from this sole uncertain parameter. A value of 1 indicates that the entire variance of the projections is explained by this sole uncertain parameter and a value of 0 indicates that the uncertain parameter has no impact on the projection uncertainty. We estimate these Sobol indices using Monte Carlo sampling following the estimation method by Janon et al. (2014). We use a total of $\nu = 4 \times 5000$ evaluations of the computational model to estimate the Sobol indices.

Figure 13 shows the Sobol sensitivity indices for each gate. Though the sum of all three sensitivity indices should be less than 1, the sum of the estimated indices exceeds this value for a few gates (gates 6, 12, and 13). This suggests that the Sobol estimates are not totally converged for some gates. From Fig. 13, we find that the uncertainty on the ice thickness has the largest influence on the uncertainty in the mass flux across fluxgates, accounting for more than 60% to 80% of their variance. The second most influential input parameter is the basal friction coefficient (Sobol indices ranging from 5% to 35%), except
for gates 1 and 13 lying on the ice shelf where there is no basal friction.

### 4.6   Transient analysis

Finally, we present a transient experiment of the Pine Island Glacier over a 10 year period. The Pine Island Glacier is forced by basal melting underneath the floating portion of the domain and surface mass balance. We impose a basal melting rate of $25\,\mathrm{m\,yr^{-1}}$ and we consider the surface mass balance (smb) to be uncertain. We represent the uncertain smb as

$$\mathrm{smb}(\boldsymbol{s}, t) = \mathrm{smb}_{\mathrm{ref}}(\boldsymbol{s})(1 + \sigma_{\mathrm{smb}} p_{\mathrm{smb}}(\boldsymbol{s}, t)), \tag{23}$$





where smb$_\text{ref}$ is a reference value for the smb taken from Vaughan et al. (1999), $\sigma_\text{smb}$ a constant 10%-error margin, and $p_\text{smb}(\boldsymbol{s}, t)$ a spatio-temporal random perturbation. The spatio-temporal random perturbation is represented as a spatio-temporal Gaussian random field given by Eq. (17). The spatial noise has a parameter range of 30 km. We assume that the smb exhibits a temporal monthly variability and consequently we discretize the temporal process in time with a one-month time step. We use the same

time step to run the ISSM model forward in time. Figures 14 and 15 show two realizations of the random perturbation for a temporal correlation of 0.5 and 0.95, respectively.

Figure 16 shows the statistical distribution of mass flux through gate 3 as a function of time for a temporal correlation of 0.5. We observe an increase in the mean mass flux through the gate due mainly to the ocean forcing, with the estimated mean values approximately equal to the mass flux in the absence of stochastic smb forcing. Most gates show an increase in their

mean mass flux in time expect for gate 4 that shows a decrease in its mean mass flux and gates 2, 8, 9, and 13 that show a relatively constant mean mass flux in time. However, we observe for all gates an increase in the coefficient of variation of the mass flux, thus suggesting that mass balance estimates become increasingly uncertain in time.

Figure 17 shows for a few gates the coefficient of variation at the end of the simulation as a function of the temporal correlation $\phi$. From this figure, it is clear that increased temporal correlation results in increased uncertainty in the mass flux,

with a more significant increase for higher values of the temporal correlation.

## 5  Discussion

Model-based projections of ice-sheet mass balance should all be ideally given in the form of probabilistic projections that provide an assessment of the impact of uncertainties in boundary conditions, climate forcing, ice sheet geometry, or initial conditions. In this context, uncertainty quantification analysis with ice-sheet models should capture the spatial and temporal

variability of these sources of uncertainties. To this end, random fields are an essential statistical tool for the modeling and analysis of uncertain spatio-temporal processes. Here, we take advantage of the explicit link between Gaussian random fields with Matérn covariance function and a stochastic partial differential equation to implement a new sampling capability within ISSM. The FEM-oriented implementation of ISSM and its fully object-oriented and highly parallelized architecture allow for a straightforward and computationally efficient implementation of this SPDE. This new sampling capability provides an

alternative to the pre-existing UQ mesh partitioning approach implemented within the ISSM-DAKOTA UQ framework. The partitioning approach can be interpreted as the representation of a random field with perfect correlation for locations in a same partition and zero (Larour et al., 2012b) or non-zero correlation (Larour et al., 2020) for locations in different partitions. Because the partitions are constant throughout the sampling, realizations strongly depend on the partitioning, which is not the case with the new sampling capability we implemented. In the partitioning approach, the number of partitions artificially

controls the spatial scale of the random perturbation. In contrast, spatial correlation is an intrinsic feature of Gaussian Matérn random fields and can be controlled through the parameter range $\rho$. In addition, random samples generated following the SPDE approach are mesh independent as long as the mesh resolution is sufficiently fine to capture the desired spatial variability.





The results presented here provide complementary insight into the impact of uncertainties on the PIG. First, our results are consistent with previous results by Larour et al. (2012b) showing that the SSA model is stable to uncertainties in ice thickness. For most gates, the probability distribution of the mass flux can be considered as a Gaussian distribution centered around the reference value. The overall uncertainty is relatively small and does not exceed at most a few percents of the mean value. This suggests that the response of the PIG to changes in ice thickness is relatively smooth even if the ice flow dynamics is nonlinear. Therefore, robust projections of mass balance estimates of the PIG are possible even in the presence of uncertainties in ice thickness. Second, our results show that a higher spatial correlation length results in larger uncertainty in mass balance estimates. The correlation length controls the size of the region around each gate in which changes in ice thickness significantly impact mass flux through the gates (see Fig. 12). This conclusion is consistent with Schlegel et al. (2018) who showed that many small partitions would likely underestimate uncertainty in model outputs while the use of a single partition would likely overestimate the uncertainty. Third, our sensitivity analysis shows that ice thickness is the most influential parameter in inducing uncertainty in mass flux estimates. Although this conclusion is consistent with Larour et al. (2012b), both conclusions cannot be directly compared because we carried out a global sensitivity analysis rather than a local sensitivity analysis as performed in the latter study. A local sensitivity analysis measures the sensitivity of the mass flux to local changes in input data and assumes that each input variable is independent. This hypothesis is not appropriate for correlated random fields. In contrast, the global sensitivity analysis in Sect. 4.5 allows to consider the global uncertainty in the random field and only requires the different random fields to be statistically independent. Fourth, our transient simulation shows that uncertainty in mass balance estimates increases in time and for higher temporal correlations.

Finally, we discuss a few limitations of the SPDE approach. First, this approach relies on the assumption that uncertainty in spatially varying input parameters can be appropriately characterized by a Gaussian random field. Though Gaussian random fields are a practical model for many stochastic phenomena, this assumption may not be appropriate because input parameters can have an asymmetric distribution, heavy tails, or simply be positive or bounded. While this may sound restrictive, non-Gaussian random fields can be obtained through nonlinear transformations of Gaussian random fields (Vio et al., 2001; Hristopulos, 2020) or by solving the SPDE with a non-Gaussian white noise (Bolin, 2014). Second, representing spatio-temporal processes with a separable autoregressive model relies on the assumption that the spatial and temporal variabilities are uncoupled and that the temporal variability can be simply described with a first-order Markov process. These two assumptions may be too restrictive to describe the complex spatio-temporal variability of stochastic processes like the surface mass balance (White et al., 2019). Third, the SPDE and autoregressive models require the specification of model parameters, most notably the spatial and temporal correlation lengths. As shown in Sect. 4, these parameters have a major impact on the uncertainty in mass balance estimates. Therefore, these parameters should ideally be constrained through data assimilation methods, modeling efforts, and inverse methods, including Bayesian inference; see, for instance Cameletti et al. (2012) and Western et al. (2020), for Bayesian spatio-temporal inference with hidden Gaussian random fields. Lastly, we assumed the spatial correlation to be constant on the PIG. While this assumption might seem reasonable for a single glacier, this may be more questionable for an ice sheet with different drainage basins. However, this issue may be handled by considering a spatially varying parameter $\kappa$ as already discussed in Sect. 2.1.1.





# 6 Conclusions

We presented and implemented a new sampling capability within ISSM for uncertainty quantification analysis of spatially
(and temporally) varying uncertain input parameters in ice-sheet models. So far, sampling of spatially varying uncertain in-
put parameters within ISSM has relied on a partitioning of the computational domain and sampling from the partitions. This
approach may be limited in representing spatial and temporal correlations. To improve the probabilistic characterization of spa-
tially (and temporally) correlated uncertain input parameters, we proposed to generate realizations of Gaussian random fields
with Matérn covariance function. Our implementation relies on an explicit link between Gaussian random fields with Matérn
covariance function and a stochastic partial differential equation. This SPDE sampling approach allows for a computationally
efficient sampling of random fields using the FEM and provides a flexible mathematical framework to build probabilistic mod-
els of spatio-temporal uncertain processes. In addition, model parameters allow to control intrinsically the spatial and temporal
correlations of the realizations as well as their variance and smoothness.

We applied this new capability to investigate the impact of spatially (and temporally) varying sources of uncertainties,
namely uncertainties in ice thickness, basal friction, ice hardness, and surface mass balance, on the Pine Island Glacier. More
specifically, we investigated the impact of spatial and temporal correlations on ice-sheet mass balance. We showed that the SSA
model is stable to uncertainties. We also showed that larger correlation lengths lead to increased uncertainty in mass balance
estimates because the random perturbations have a larger length scale of influence. We found that the most influential source
of uncertainties for estimating mass balance is the uncertainty in ice thickness. We also showed in a transient experiment that
uncertainty in mass balance estimates increases in time and with higher temporal correlation. Overall, our results demonstrate
the need to better constrain the spatial and temporal variability of physical processes impacting ice-sheet dynamics through
data assimilation and modeling efforts.

*Code and data availability.* The ISSM code can be downloaded, compiled, and executed following the instructions available on the ISSM
website https://issm.jpl.nasa.gov/download (last access: 7 September 2021). The public SVN repository for the ISSM code can also be found
directly at https://issm.ess.uci.edu/svn/issm/issm/trunk and downloaded using username "anon" and password "anon". The archived version
of the source code used in this manuscript is made available as part of a Zenodo repository at https://doi.org/10.5281/zenodo.5532775. Data
to reproduce the results in Sects. 3 and 4 are made available as part of a Zenodo repository at https://doi.org/10.5281/zenodo.5532710.

*Author contributions.* Both authors discussed the results presented in this paper. K.B. implemented the sampling capability into ISSM and
carried out the simulations. He wrote the bulk of the manuscript with relevant comments from E. L.

*Competing interests.* The authors declare that they have no conflict of interest.





*Acknowledgements.* K.B. acknowledges the support from the NASA Postdoctoral Program (NPP), administered by the Universities Space Research Association (USRA) under contract with the National Aeronautics and Space Administration (NASA). The research described in this manuscript was carried out at the Jet Propulsion Laboratory, California Institute of Technology, under a contract with the NASA. Government sponsorship is acknowledged.





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





**Figure 10.** Normalized histograms for mass flux across fluxgate 3 (see Figure 5) for different parameter ranges. The mean $\mu$ (in Gt yr$^{-1}$) and the coefficient of covariation are provided for each parameter range. The kernel density estimate of the probability density function is shown in red and its Gaussian approximation with corresponding mean and coefficient of variation in green.



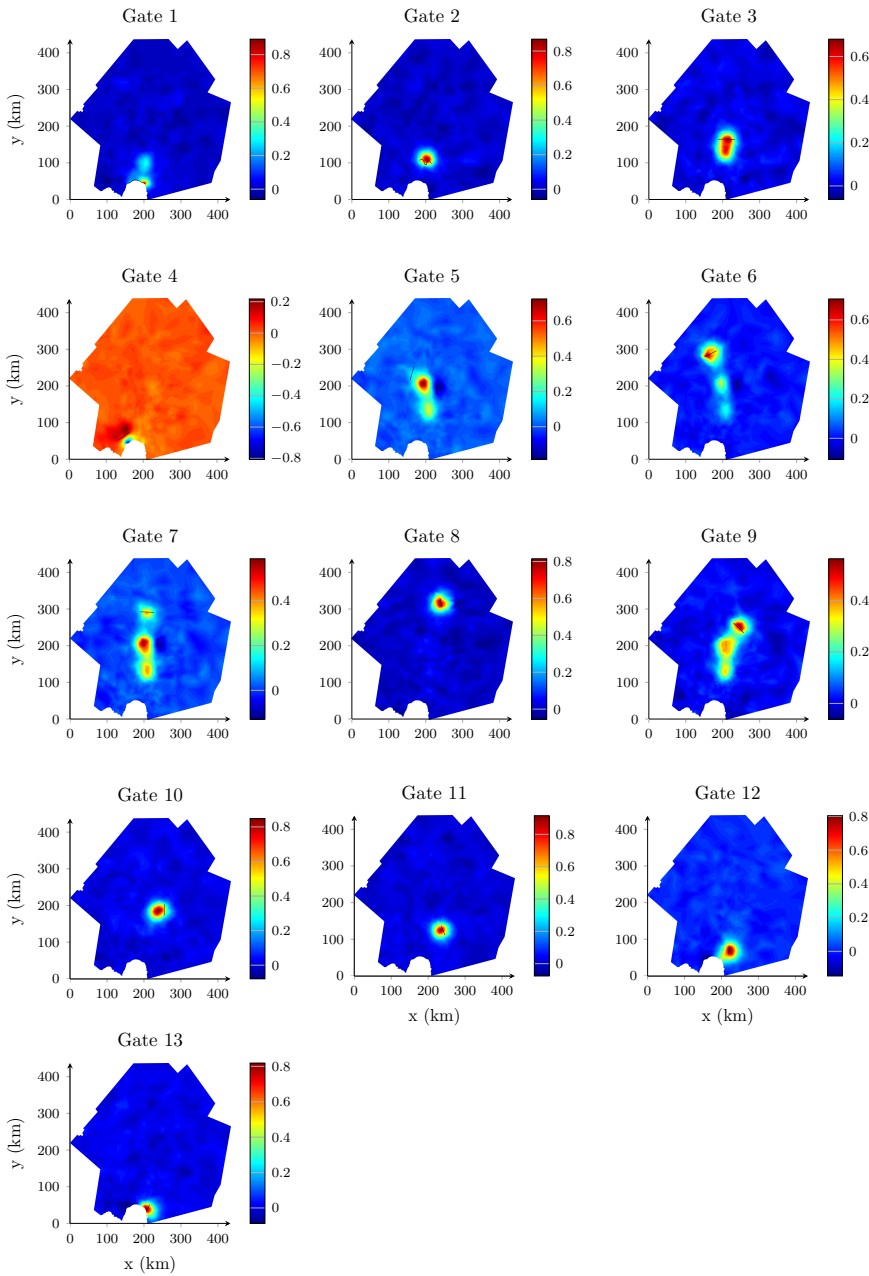

**Figure 11.** Sensitivity map (visual representation of the coefficient of correlation) for mass flux across fluxgates depicted in Figure 5. Results are for a parameter range of 30 km.



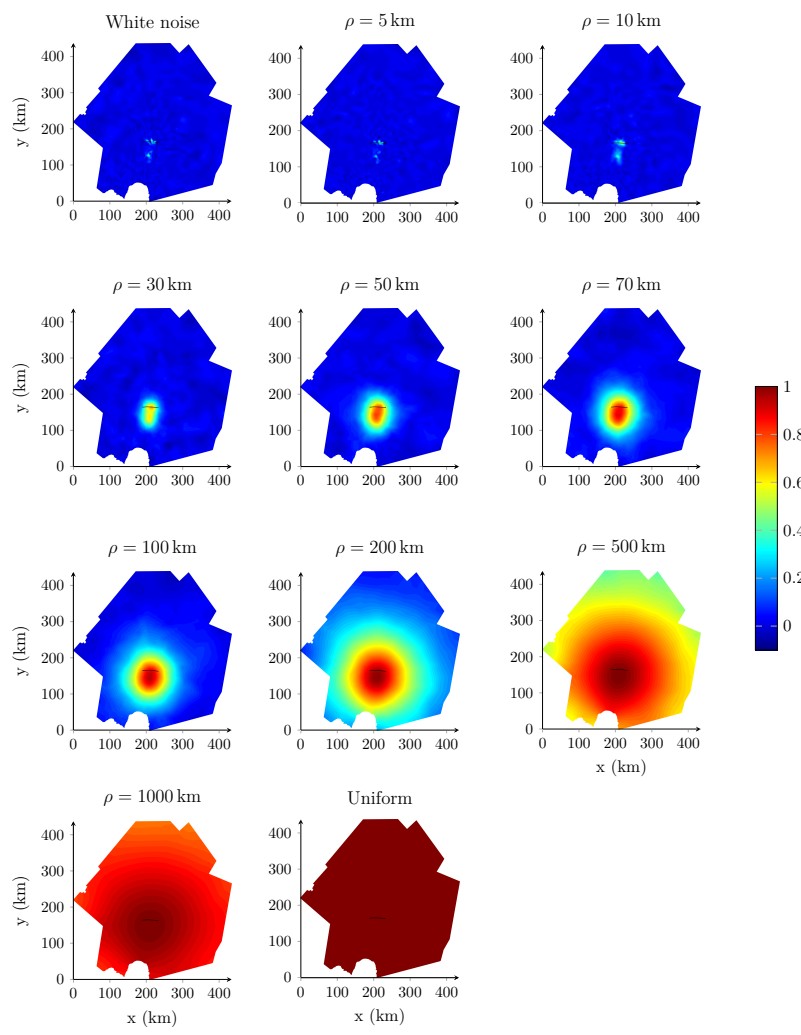

**Figure 12.** Sensitivity map (visual representation of the coefficient of correlation) for mass flux across fluxgate 3 (see Figure 5) for different parameter ranges.



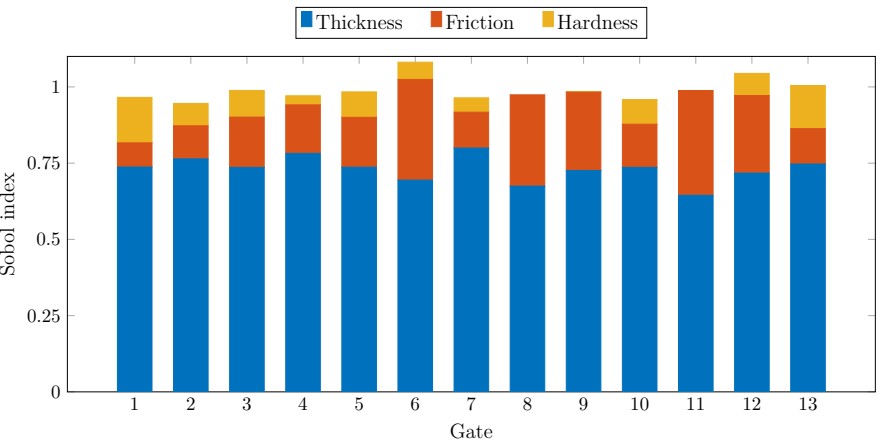

**Figure 13.** Sobol sensitivity indices for the mass flux across flux gates. The gap between the height of a bar and the unit value represents the interaction index.



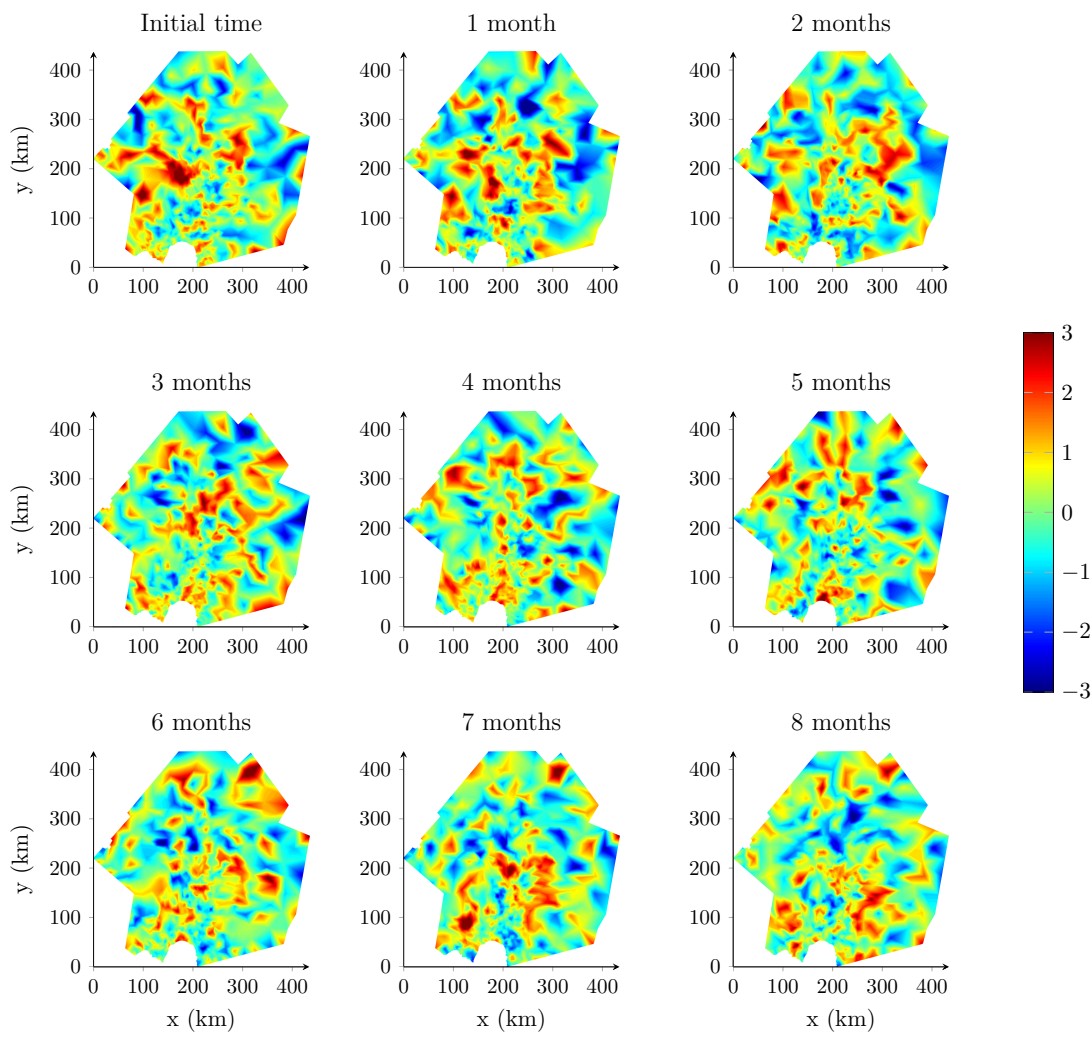

**Figure 14.** Transient samples of random perturbation over Pine Island Glacier with a temporal correlation of 0.5.





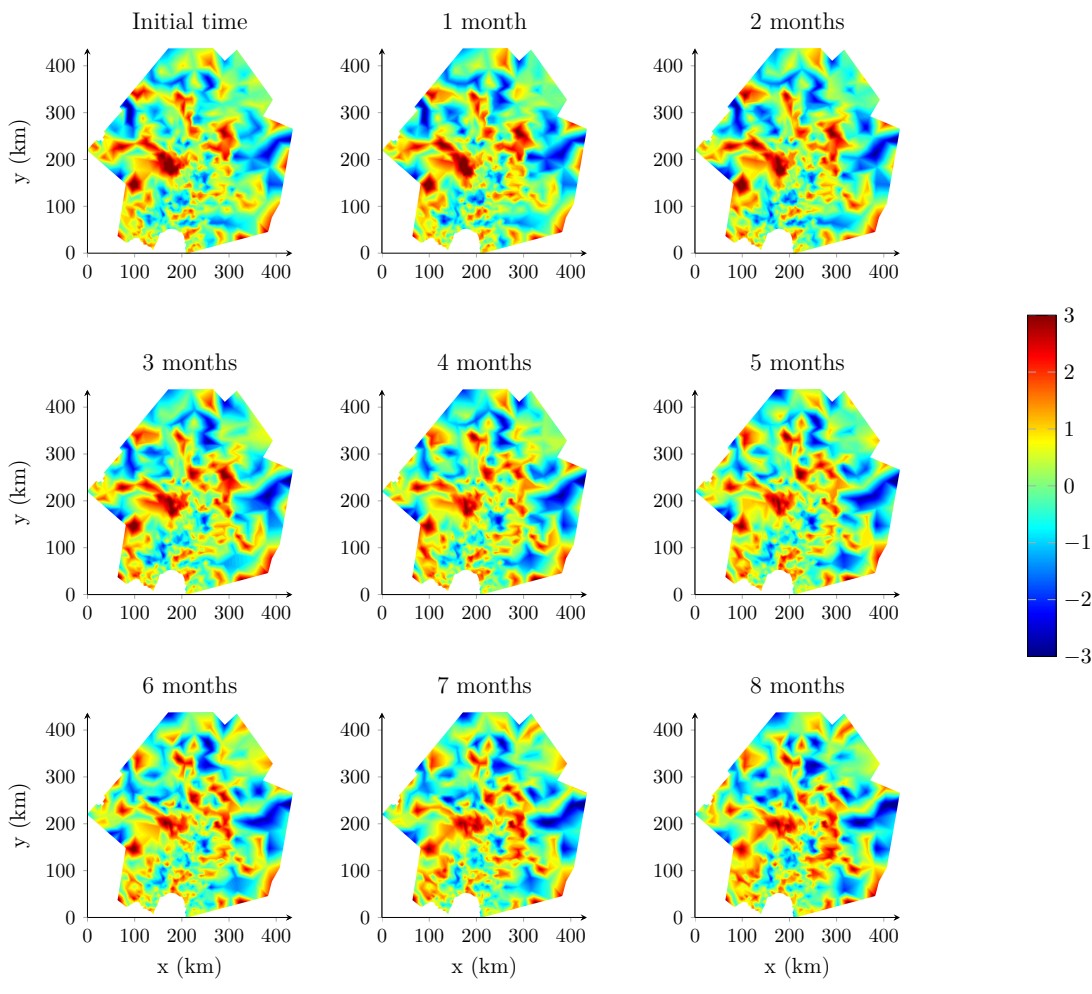

**Figure 15.** Same as Fig. 14 but with a temporal correlation of 0.95. Figures 14 and 15 have the state initial state.







**Figure 16.** Normalized histograms for mass flux across fluxgate 3 (see Figure 5) as a function of time. The mean (in Gt/yr) and the coefficient of variation (in percent) are provided for each time. The kernel density estimate of the probability density function is shown in red and its Gaussian approximation with corresponding mean and coefficient of variation in green. Results are for a parameter range of 30 km and a temporal correlation of 0.5.





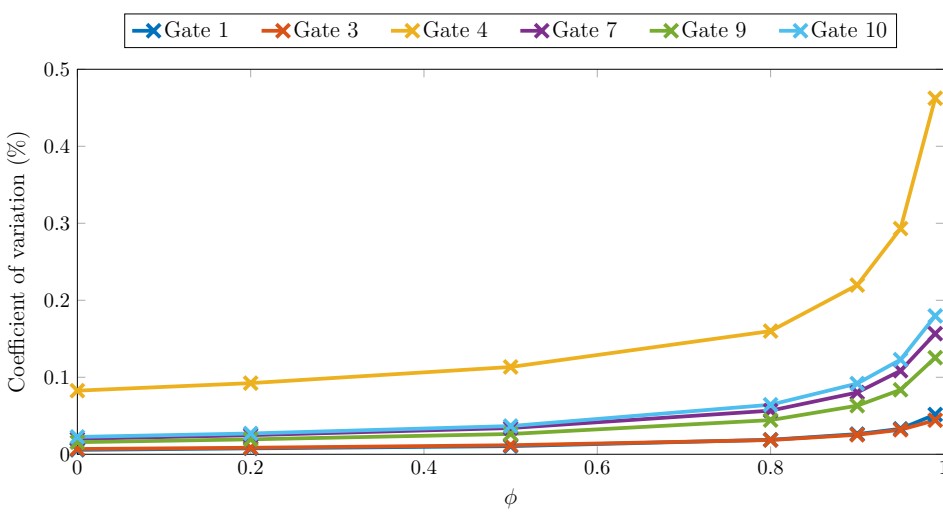

**Figure 17.** Coefficient of variation at the end of the simulation as a function of the temporal correlation $\phi$. Results are for different gates.