# Peer review of "Implementation of a Gaussian Markov random field sampler for forward uncertainty quantification in the Ice-sheet and Sea-level System Model v4.19"

_Geoscientific Model Development, 2021_

## Referee Comment (RC1)

**Referee Comment on 'A new sampling capability for uncertainty quantification in the Ice-sheet and Sea-level System Model v4.19 using Gaussian Markov random fields'**

October 2021

**1  Summary**

In this manuscript, the authors implement a method by which samples from a Gaussian Random Field with Matern covariance can be drawn. This method is based on the stochastic PDE approach. Using this sampling mechanism, the manuscript shows a handful of experiments demonstrating the sensitivity of mass flux to input fields (e.g. thickness, traction) perturbed according to this random noise. Tha manuscript also demonstrates an autoregressive approach to generating time-correlated noise, and shows the experimental distribution of mass loss due to an uncertain surface mass balance perturbed in this way.

The methods presented here are already mature, both from the perspective of the sampling technique and the ice sheet model. There's no data assimilation or attempts to tune the hyperparameters of the sampling method, so there's nothing to discuss with respect to inference. The patterns of sensitivity are about as one would expect. It's good that this capability exists, and will be useful in future studies. The paper is mostly written clearly, and does a decent job of placing this work in the appropriate context. As such, I only have some minor technical points, discussed below. However, this brevity is because there just isn't much scientific impact presented to comment on.

**Minor Points**

**78** This WLOG statement isn't true: of course there's a loss of generality from assuming a mean of zero. However, it's reasonable to argue that it doesn't matter because you later plan to use these as relative perturbations to some a priori inferred mean.

**106** It's worth noting here that Gaussian random fields are already being employed in Isaac, 2015 (already referenced in this paper at another location)

and also in Brinkerhoff, 2021 ( https://arxiv.org/abs/2108.07263 ), both of which use low-rank approximations to yield the problem tractable.

**144** It's worth mentioning the downsides to the SPDE approach as well: in the inverse context, it provides no immediate solution for how to represent posterior covariance.

**160** It's not immediately obvious that the matrix square root (particularly of $K$) should be easy to compute, or that it should retain sparsity. If it doesn't retain sparsity, then the scalability of this method could be substantially limited.

**Eq. 16** Is there a more rigorous means to quantify what potential errors that this mass lumping step induces?

**192** I don't understand this paragraph. if $\phi = 1$ then shouldn't nothing change at all (contrary to the statement that it leads to a random walk)? It seems to me that the equations would yield

$$x_t = x_t + \epsilon_t$$

$$\epsilon_t \sim \mathcal{N}(\mu = 0, \sigma^2 = 0)$$

.

I may be misunderstanding, but perhaps clarification would be helpful here.

**285** Can 'converged' be rigorously defined here?

**Fig. 16** Might this be better represented as a single plot, with $t$ as the independent-variable?

---

## Author Comment (AC1)

***Response to the Referee comment on*** "**A new sampling capability for uncertainty quantification in the Ice-sheet and Sea-level System Model v4.19 using Gaussian Markov random fields**" ***by*** **Kevin Bulthuis and Eric Larour**

**1 Summary statement**

In this manuscript, the authors implement a method by which samples from a Gaussian Random Field with Matern covariance can be drawn. This method is based on the stochastic PDE approach. Using this sampling mechanism, the manuscript shows a handful of experiments demonstrating the sensitivity of mass flux to input fields (e.g. thickness, traction) perturbed according to this random noise. Tha manuscript also demonstrates an autoregressive approach to generating time-correlated noise, and shows the experimental distribution of mass loss due to an uncertain surface mass balance perturbed in this way.

The methods presented here are already mature, both from the perspective of the sampling technique and the ice sheet model. There's no data assimilation or attempts to tune the hyperparameters of the sampling methods, so there's nothing to discuss with respect to inference. The patterns of sensitivity are about as one would expect. It's good that this capability exists, and will be useful in future studies. The paper is mostly written clearly, and does a decent job in placing this work in the appropriate context. As such, I only have some minor technical points, discussed below. However, this brevity is because there just isn't much scientific impact presented to comment on.

We would like to thank anonymous referee #1 for the time dedicated to this manuscript and his/her constructive comments to improve the general quality and readibility of the manuscript. We will try to give a proper response to his/her comments. For each referee's comment (written in blue), we included below a response (written in black) and proposed means to improve the manuscript.

**2 Minors comments**

l.78: This WLOG statement isn't true: of course, there's a loss of generality from assuming a mean of zero. However, it's reasonable to argue that it doesn't matter because your later plan to use these as relative perturbations to some a priori inferred mean.

Agree. We remove the WLOG statement.

- l.106: It's worth noting here that Gaussian random fields are already being employed in Isaac, 2015 (already referenced in this paper at another location) and also in Brinkerhoff, 2021( https://arxiv.org/abs/2108.07263 ), both of which use low-rank approximations to yield the problem tractable.

Thank you for the suggestion. We have added these references as well as Babaniyi et al., 2021 (https://tc.copernicus.org/articles/15/1731/2021/) as references for applications of Gaussian random fields in glaciology.

We have added the following sentence: "Gaussian random fields have already been employed in glaciology in a number of studies including Isaac et al. (2015), Babaniyi et al. (2021) and Brinkerhoff (2021)."

- l.144: It's worth mentioning the downsides to the SPDE approach as well: in the inverse context, it provides no immediate solution for how to represent posterior covariance.

Thank you for the suggestion. We recognize that the manuscript lacks a bit of perspective about the SPDE approach in the important context of inverse problems and that it cannot be used directly to represent posterior covariance.

We have added the following paragraph in the manuscript:" The SPDE approach can also be used to define a proper choice of a prior distribution for inverse problems in infinite dimension (Bui-Thanh et al., 2013; Isaac et al., 2015; Petra et al., 2014; Stuart, 2010). However, this approach does not provide any immediate solution to represent the posterior covariance. For general inverse problems, the posterior distribution does not need to be Gaussian even if the prior distribution is a Gaussian random field. In this case, the posterior covariance can be estimated using, for instance, Markov chain Monte Carlo algorithms (Beskos et al., 2017; Petra et al., 2014) or a Laplace approximation of the posterior distribution (Bui-Thanh et al., 2013; Isaac et al., 2015).

- l.160: It's not immediately obvious that the matrix square root (particularly of $K$) should be easy to compute, or that it should retain sparsity. If it doesn't retain sparsity, then the scalability of this method could be substantially limited.

Indeed, there is a priori no reason for the square root of the matrices $\mathbf{K}$ and $\mathbf{M}$ to retain the sparsity of both matrices. The computation of these square root matrices represent an important computational cost compared to other numerical operations required by the SPDE approach. This motivates the use of a lump matrix approximation of the mass matrix. A lump matrix approximation of the matrix $\mathbf{K}$ can also be considered for $\alpha = 1$ (and odd values of $\alpha$) in order to compute the $\mathbf{K}^{1/2}$. It should be better acknowledged in our manuscript.

We have changed the sentence as: "The bulk of the computational cost is in evaluating the square root of the matrix $\mathbf{M}$ or $\mathbf{K}$. Even if the matrices $\mathbf{M}$ and $\mathbf{K}$ are sparse, their square root does not need to be sparse. In order to speed up the computation and retain sparsity, the mass matrix $\mathbf{M}$ (idem for $\mathbf{K}$ if $\mathbf{K}^{1/2}$ needs to be evaluated) can be approximated as a diagonal lump mass matrix $\widetilde{\mathbf{M}}$"

- Eq. 16: Is there a more rigorous means to quantify what potential errors that this mass lumping step induces.

While we think that investigating the impact of the mass lumping (or Markov) approximation is beyond the scope of this paper, we want to mention that it has been studied in Appendix C5 in Lindgren et al. (2011). Following Lindgren et al. (2011) the convergence rate for the Markov approximation is the same as for the full finite-element model. Bolin and Lindgren (2009) have also shown negligible differences between the exact finite-element model representation and the Markov approximation.

We have added the following sentence to the manuscript for further references regarding the mass lumping approximation: "The Markov approximation of the Gaussian random field has been shown to have negligible differences with the exact finite element representation (Bolin and Lindgren, 2009) and its convergence rate is the same as the exact finite element representation (Lindgren, 2011).

- l.192: I don't understand this paragraph. if $\phi = 1$ then shouldn't nothing change at all

(contrary to the statement that it leads to a random walk)? It seems to me that the equations would yield,

$$x_t = x_t + \epsilon_t, \tag{1}$$

$$\epsilon_t \sim \mathcal{N}(\mu = 0, \sigma^2 = 0) \tag{2}$$

I may be misunderstanding, but perhaps clarification would be helpful here.

Indeed, the paragraph may be a bit confusing. The variance of an AR1 process can be computed as

$$\mathbb{V}[x_t] = \phi^2 \mathbb{V}[x_{t-1}] + \mathbb{V}[\epsilon_t] = \phi^2 \mathbb{V}[x_{t-1}] + \sigma_\epsilon^2. \tag{3}$$

The condition on the variance of the noise for the autoregressive to be stationary in time with constant variance $\sigma^2$ is that $\sigma_\epsilon^2 = \sigma^2(1-\phi^2)$. If $\phi = 1$, it requires indeed the noise term to have a zero variance for the autoregressive model to be stationary in time. We did not consider the degenerate case of a Gaussian noise with zero variance when writing this paragraph (as there would be no randomness in the process), but this should be made more explicit to avoid any confusion. If we impose $\sigma_\epsilon^2 > 0$, then the autoregressive model is indeed a random walk for $\phi = 1$.

To avoid any confusion, we change the paragraph as follows: "At every time step $t$, the noise term $\epsilon_t(\boldsymbol{s})$ is chosen as a Gaussian random field with Matérn covariance function and positive variance $\sigma_\epsilon^2$, obtained as the solution of the SPDE (3). If $|\phi| < 1$, $x_t$ is a stationary process in time with zero mean and marginal variance $\sigma^2$ if $x_0$ is a Gaussian random field with zero mean and variance $\sigma^2$ and the noise variance is chosen as $\sigma_\epsilon^2 = \sigma^2(1-\phi^2)$. If $|\phi| \geq 1$, the process is non stationary in time, with the case $\phi = 1$ corresponding to a discrete random walk."

- l.285: Can 'converged' be rigorously defined here.

Indeed the statement 'reasonable convergence' might seem a little bit vague. We estimated the estimation error for the mean and standard deviation of the mass flux via bootstrapping. The bootstrap error is of a few hundredths of percent for the mean value and a few percents for the standard deviation. We have indicated these values at the end of the sentence.

- Fig. 16 Might this be better represented as a single plot, with t as the independent variable?

We thank the referee for his/her suggestion. Such a single plot would definitely make sense. Unfortunately, the uncertainty in the mass flux estimates is so tiny that it cannot be represented properly on a single plot as a function of time. Because we want to highlight the increase in the uncertainty over time, we find the current figure more appropriate for our purpose.

**3 References**

Babaniyi, O., Nicholson, R., Villa, U., and Petra, N. Inferring the basal sliding coefficient field for the Stokes ice sheet model under rheological uncertainty, The Cryosphere, 15, 1731-1750, https://doi.org/10.5194/tc-15-1731-2021, 2021.

Beskos, A., Girolami, M., Lan, S., Farrell, P. E., and Stuart, A. M.: Geometric MCMC for infinite-dimensional inverse problems, J. Comput. Phys., 335, 327–351, https://doi.org/10.1016/j.jcp.2016.12.041, 2017.

Bolin, D. and F. Lindgren, F.. Wavelet Markov models as efficient alternatives to tapering and convolution fields. Technical report, Mathematical Statistics, Centre for Mathematical Sciences, Faculty of Engineering, Lund University, 2009.

Brinkerhoff, D. J. Variational Inference at Glacier Scale, arXiv, https://arxiv.org/abs/2108.07263, 2021.

Bui-Thanh, T., Ghattas, O., Martin, J., and Stadler, G.: A Computational Framework for Infinite-Dimensional Bayesian Inverse Prob- lems Part I: The Linearized Case, with Application to Global Seismic Inversion, SIAM J. Sci. Comput., 35, A2494–A2523, https://doi.org/10.1137/12089586x, 2013.

Isaac, T., Petra, N., Stadler, G., and Ghattas, O. Scalable and efficient algorithms for the propagation of uncertainty from data through inference to prediction for large-scale problems, with application to flow of the Antarctic ice sheet, J. Comput. Phys., 296, 348–368, https://doi.org/10.1016/j.jcp.2015.04.047, 2015.

Bolin, D. and Lindgren, F.: Excursion and contour uncertainty regions for latent Gaussian models, J. Royal Stat. Soc. B, 77, 85–106, https://doi.org/10.1111/rssb.12055, 2015.

Lindgren, F., Rue, H., and Lindström, J. An explicit link between Gaussian fields and Gaussian Markov random fields: the stochastic partial differential equation approach, J. R. Stat. Soc. B, 73, 423-498, https://doi.org/10.1111/j.1467-9868.2011.00777.x, 2011.

Petra, N., Martin, J., Stadler, G., and Ghattas, O.: A Computational Framework for Infinite-Dimensional Bayesian Inverse Problems, Part II: Stochastic Newton MCMC with Application to Ice Sheet Flow Inverse Problems, SIAM J. Sci. Comput., 36, A1525–A1555, https://doi.org/10.1137/130934805, 2014.

Stuart, A. M.: Inverse problems: A Bayesian perspective, Acta Numer., 19, 451–559, https://doi.org/10.1017/s0962492910000061, 2010.

---

## Author Comment (AC2)

***Response to the Referee #2 comment on* "A new sampling capability for uncertainty quantification in the Ice-sheet and Sea-level System Model v4.19 using Gaussian Markov random fields" *by* Kevin Bulthuis and Eric Larour**

The authors added support for Gaussian random fields with Matern-type covariance functions in ISSM (Ice-sheet and Sea-level System Model) and in this paper, they show the resulting new capabilities ISSM offers for forward UQ. The authors describe how to draw samples from a Gaussian distribution with Matern covariance and use these samples to run forward UQ for a few ice sheet flow models.

The topic of the paper is interesting. In terms of novelty as mentioned above, the mathematical techniques described in this paper are all well-known, hence there is not much one can comment on this aspect. The validation tests ("sanity checks") are nice and may be useful, especially to the users of ISSM. Below I list a few comments/concerns:

1. The examples are interesting but lack sufficient details. For instance, there are several things explained in words but there is no concrete problem/mathematical description, only several references to previous work are provided. The authors spent significant effort to explain the sampling procedure, which is known, but unfortunately rushed through the numerical experiments.

   We agree with the referee with the fact that the description of the problem may seem succinct, especially for people not familiar with ice-sheet modeling. However, the focus of this paper is on the sampling procedure (even though it is known for people in UQ) rather than the description of the ice-sheet dynamics, which we believe is not required to understand the numerical experiments. We will be pleased to clarify some points on the numerical experiments if the referee has any specific question on them.

2. How are the parameters (mean, correlation, etc.) chosen for the Gaussian distributions one samples from for the forward UQ? Without a proper data assimilation or inversion process for these ice sheet problems, not sure how realistic these distributions are and certainly not sure how much these can be trusted for prediction and UQ.

   The mean values for the Gaussian distributions are based on reference values derived from bedrock and surface elevation data for the ice thickness, a thermal model and surface temperature data for the ice hardness and an inverse method for the basal drag. Regarding the choice of the parameters of the SPDE, our goal was not to provide realistic distributions or predictions but rather to illustrate how the implemented sampler can be used for forward UQ analysis. For this reason, the parameter range (30 km) has been chosen rather arbitrarily but is believed to represent supposed variations in ice thickness. To assess the impact of the parameter range, we also provided results for other parameter ranges. In order to provide trusty predictions, these parameters should indeed ideally be constrained with a proper data assimilation or inversion process. Such methods are however beyond the scope of this manuscript but we plan to constrain these parameters in a future work.

   We have added the following sentences at the end of Section 4.2: For this application, we have chosen the values of the parameter range arbitrarily without any proper data assimilation or inversion process. The use of an unconstrained parameter range is justified by the fact that we do not seek to provide new probabilistic mass-balance estimates for the Pine Island Glacier but seek only to demonstrate the interest of the implemented stochastic sampler for forward UQ analysis.

3. What was the dimension of the unknowns or quantity of interest? Are the sampling and forward UQ processes described scalable and computationally tractable for large-scale ice sheet problems?

We thank the referee for this question. Indeed, the paper lacks some details regarding the dimension (or size) of the problem. For the problem considered in Section 4, the mesh is made of 2085 elements and 1112 nodes, which corresponds to a relatively small dimension compared to other ice-sheet problems. For our purpose, a higher spatial resolution was not required. Therefore, the dimension of the discretized random field is of 1112. The quantity of interest is of dimension 13 (for the mass fluxes through the 13 gates), even though the calculation of each mass flux requires the prior calculation of the ice thickness and the velocity field. In our example, the cost of the sampler is relatively negligible (only a few tenths of seconds per sample). The cost of solving the ice-sheet model is of a few seconds per run. We have added these values to the manuscript.

Regarding the scalability of the method, the sampler scales well for larger ice-sheet problems. The SPDE approach in itself is highly scalable and efficient for large-scale problems (see for instance Isaac et al. (2015)). The computational cost of the sampler is in any case (much) lower that the cost of solving the ice-sheet model. Therefore, the cost for the forward UQ analysis is determined by the computational cost of the ice-sheet solver, which can indeed be computationally prohibitive if the dimension of the problem at hand is too high (especially for transient simulations). However due to the nature of the Monte Carlo simulations, all simulations can be run in an embarrassingly parallel way. If computational resources are limited, then other methods for UQ propagation like advanced Monte Carlo sampling approaches or surrogate modeling might be of interest to be investigated.

4. How exactly the convergence of the samples is assessed?

This question has also been raised by referee#1. The statement 'reasonable convergence' might seem a little bit vague. We can estimate the estimation error for the mean and standard deviation of the mass flux via bootstrapping. The bootstrap error is of a few hundredths of percent for the mean value and a few percents for the standard deviation. We have indicated these values at the end of line 285.

5. It is unclear what the novelty in this paper is. Is the goal to present these new capabilities ISSM provides and compare the new results with previous forward UQ studies? If so, the abstract of the paper is a bit misleading. This suggests that the authors propose a new sampling technique for UQ.

Indeed, the novelty of this paper is not in the sampling technique in itself, which is already well-established in UQ and has been used in a number of studies including applications in glaciology (though with a different goal). In particular, the use of the words "new sampling capability" might be a bit misleading and might suggest that we are actually proposing a new sampling technique in uncertainty quantification. The goal of this paper is more to introduce a (new) stochastic sampler for Gaussian Matérn random fields within ISSM and to illustrate the interest of this sampler for forward UQ analysis in ice-sheet models. While the sampling method is well-established in UQ, we thought this paper is of interest for ice-sheet modelers who are less familiar with uncertainty quantification. We have modified the abstract and replaced "(new) sampling capability" by "stochastic sampler" in the main text in order to make the manuscript more consistent with this goal.

6. Also, from the title (and the abstract), it is not clear that only forward UQ is being considered.

   Indeed, the title may be a bit misleading. To highlight the fact that only forward UQ is being considered and that we are not proposing a new sampling technique (but only implementing an existing one), we have changed the title of this paper as "Implementation of a Gaussian Markov random field sampler for forward uncertainty quantification in the Ice-sheet and Sea-level System Model v4.19".

**References**

Isaac, T., Petra, N., Stadler, G., and Ghattas, O. Scalable and efficient algorithms for the propagation of uncertainty from data through inference to prediction for large-scale problems, with application to flow of the Antarctic ice sheet, J. Comput. Phys., 296, 348–368, https://doi.org/10.1016/j.jcp.2015.04.047, 2015.

---

## Author Response (AR2)

***Response to the minor comments on*** "Implementation of a Gaussian Markov random field sampler for forward uncertainty quantification in the Ice-sheet and Sea-level System Model v4.19" *by* **Kevin Bulthuis and Eric Larour**

The authors did an excellent job addressing the reviewer's concerns regarding the manuscript. Here are some final (more minor comments)

We thank the referees and the editor for their last comments and have updated the manuscript based on the last minor comments.

1. It would be great if the authors would incorporate some of the references mentioned in the responses to the reviewers into the introduction. None of discussions are reflected in this section, which I think should be updated.

    We agree with the editor and the referee that the references mentioned in the responses to the reviewers should be mentioned in the introduction. We have changed the introduction to mention that Gaussian random fields have already been employed in glaciology (Isaac, 2015; Babaniyi, 2021; Brinkerhoff, 2021) while the SPDE approach has already been employed in the context of Bayesian inverse problems in glaciology (Isaac, 2015; Petra, 2014).

2. Equation (1) is not very clear. Could the authors be a bit more concrete, where do these parameters enter in the model, what the ultimate goal is with this model?

    Equation (1) is just a general model in statistics to represent a random field. It is simply used to decompose the random field into a deterministic component (its mean or trend) and a stochastic component (which is really what we are interested in in this paper). But we recognized that in the context of this paper, introducing notations (like $\mu$ and $\epsilon$) that are not used in the remainder of the paper can bring some confusion. To avoid complicating the text, we have decided to remove this equation from the paper. The introductory paragraph in Section 2 now simply writes as "In this work, we model a spatially varying uncertain input parameter as a random field $\{x(\boldsymbol{s}), \boldsymbol{s} \in D\}$ indexed over the computational domain $D$. In the following, we assume this random field to be a Gaussian random field with zero mean."

3. It would be great if the authors could give some intuition or add a reference for the uncertainty representation in Equation (20).

    The uncertainty representation in Equation (20) follows the perturbation approach in Larour et al. (2012), though not written explicitly as we did in the paper. This equation can be seen as perturbing each input quantity with a multiplicative noise proportional to a local (measurement) error margin. We have added the reference and this interpretation in the text.

**References**

Babaniyi, O., Nicholson, R., Villa, U., and Petra, N. Inferring the basal sliding coefficient field for the Stokes ice sheet model under rheological uncertainty, The Cryosphere, 15, 1731-1750, https://doi.org/10.5194/tc-15-1731-2021, 2021.

Brinkerhoff, D. J. Variational Inference at Glacier Scale, arXiv, https://arxiv.org/abs/2108.07263, 2021.

Isaac, T., Petra, N., Stadler, G., and Ghattas, O. Scalable and efficient algorithms for the propagation of uncertainty from data through inference to prediction for large-scale problems, with application to flow of the Antarctic ice sheet, J. Comput. Phys., 296, 348–368, https://doi.org/10.1016/j.jcp.2015.04.047, 2015.

Larour, E., Schiermeier, J., Rignot, E., Seroussi, H., Morlighem, M., and Paden, J.: Sensitivity Analysis of Pine Island Glacier ice flow using ISSM and DAKOTA, J. Geophys. Res. Earth Surface, 117, F02 009, https://doi.org/10.1029/2011jf002146, 2012.

Petra, N., Martin, J., Stadler, G., and Ghattas, O.: A Computational Framework for Infinite-Dimensional Bayesian Inverse Problems, Part II: Stochastic Newton MCMC with Application to Ice Sheet Flow Inverse Problems, SIAM J. Sci. Comput., 36, A1525–A1555, https://doi.org/10.1137/130934805, 2014.